# Experimental assessment of AI-based interactome mapping

Genotype-phenotype relationships are mediated through intricate networks of physical and functional interactions among macromolecules. Knowledge of the interactome is vital to understand and model genetics and cellular biology. Recent advances in accurately predicting tertiary protein structures using artificial intelligence (AI) approaches such as AlphaFold[1] have revived the vision that the protein-protein interactome might be fully predictable through computational modeling of quaternary structures. Here we present a comprehensive experimental framework to systematically assess the impact of AI-driven interactome predictions for yeast[2] and human[3]. We find that the quality of high-confidence predictions is on par with established experimental approaches. However, in proteome-wide screening, the tested AI approaches underperform in the discovery of strictly novel protein-protein interactions (PPIs) compared to experimental reference interactome maps. In particular, the yeast interactome map described here identifies >40-fold more novel PPIs than its AI counterpart. Strikingly, AlphaFold provides structural models for a substantial number of experimentally identified PPIs missed by the virtual screens. Our results suggest that, at this stage, the main contribution of AI predictions is to provide quaternary structure models for experimentally identified PPIs.

Physical interactions among proteins and other biomolecules are the basis of the functional organization of cells. Knowledge about "the interactome" is indispensable for biological discovery, from small-scale hypothesis-driven approaches to the development of large-scale models of cellular systems. Consequently, numerous approaches to map the protein–protein interactome of different organisms have been developed[4]. Despite these efforts, a complete and coherent representation of any physical interactome remains missing. This is due to technical challenges in measuring physical interactions, but also the result of different interpretations of "physical interaction", partly driven by the results obtained by different technologies. Thus, interactions may be interpreted as "membership of a macromolecular complex" and then include indirect associations or, alternatively, as direct molecular contacts between polypeptide chains that form an interface with specific bonding. These contacts can be directly affected by sequence changes due to genetic variation or mutation, and can be targeted by small molecules as in drug-target therapeutics.

Recently, the prediction of accurate tertiary structures by AlphaFold[1] has revolutionized structural biology. The subsequent development of deep-learning methods to predict quaternary structures of protein complexes, primarily AlphaFold-Multimer[5], and subsequently, AlphaFold3[6], have revitalized the hope of complete and precise computational protein–protein interaction (PPI) discovery without the costs of experiments. Moreover, the availability of models for all quaternary structures, hence of the complete "contactome"[7,8], would naturally reveal interfaces and render the question of indirect versus direct associations obsolete, while facilitating drug discovery and interpretation of genetic variation. However, despite high-profile attempts to "compute the interactome"[2,3] it is unclear how accurate and how sensitive current protein interaction predictions are.

✉e-mail: pascal.falter-braun@helmholtz-munich.de; david_hill@dfci.harvard.edu; michael_calderwood@dfci.harvard.edu; jean-claude.twizere@uliege.be; marc_vidal@dfci.harvard.edu

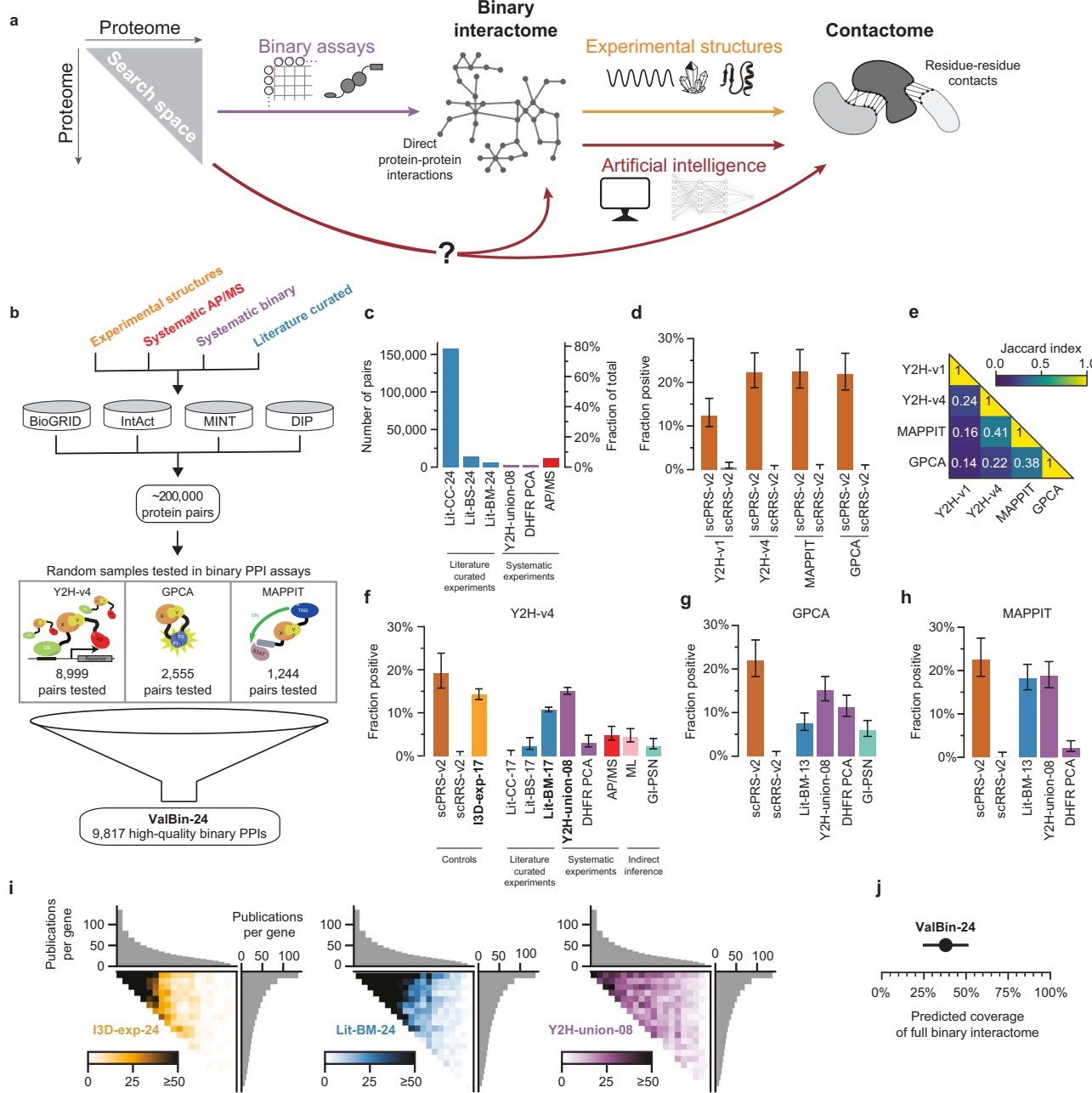

**Fig. 1 | Assessing the current state of yeast interactome mapping. a** Schematic of computational and experimental interactome mapping strategies. **b** Schematic of our approach to assess current experimental protein interaction data. **c** Division of published protein pairs into categories based on source. **d** Recovery rates of Y2H-v4 benchmarked against three other binary PPI assays, of positive and random reference sets, scPRS-v2 (108 pairs) and scRRS-v2 (198 pairs). **e** Overlap of scPRS-v2 pairs between assays. Experimental validation of various interaction datasets recovered in **f** Y2H-v4, **g** GPCA, or **h** MAPPIT. **i** Heatmaps of PPI density of the three validated binary datasets, within the proteome-by-proteome search space, ordered by the number of associated publications per gene. **j** The estimated coverage of the full interactome of the combined validated binary dataset. Error bars are 68.3% Bayesian credible intervals in (**d**, **f**–**h**) and 95% confidence intervals in (**j**).

Here, we conduct a comprehensive experimental assessment of structurally resolved interactome predictions for both *Saccharomyces cerevisiae* and *Homo sapiens*. Starting with *S. cerevisiae*, we first assessed the state of structurally resolved contactome- and interactome-mapping, focusing the latter on direct contact interactions ("binary interactions", Fig. 1a). Based on these baselines, we evaluated the quality and contribution of state-of-the-art artificial intelligence (AI) derived predictions to the yeast and human interactome maps. At their current stage of development, the most useful application of AI predictions is to complement existing experimental methods by providing quaternary structure models for novel interaction pairs.

## Results

### Over half the yeast interactome undiscovered

We started our study by assessing the state of interactome mapping in yeast. Protein interactions with experimentally resolved structures represent contactome maps with the highest level of quality and resolution, however such data exists for only 2576 heteromeric yeast PPIs between 1408 proteins (I3D-exp-24)[9] (Supplementary Data 1). This corresponds to about 8% of the interactome, and with the current growth trajectory it would take more than 100 years before a structural map of the full contactome becomes available (Supplementary Fig. 1a). Therefore, we next examined the interactome at the level of PPIs,

which we define as direct physical contacts between two proteins[4]. We found that public repositories list ~200,000 protein pairs (Fig. 1b, c), vastly in excess of the estimated interactome size of 20,000 to 40,000 PPIs[10–13]. Likely, this discrepancy is primarily driven by the inclusion of indirect associations between proteins in large complexes that are not in direct contact[14]. Eighty-three percent of the reported pairs were obtained using experimental methods related to affinity chromatography, such as affinity purification mass spectrometry (AP/MS), which, in general, cannot distinguish between co-complex indirect associations and direct PPIs[15] (Supplementary Fig. 1b). A second factor possibly inflating the number of reported interactions is low data quality. Despite the perception of literature-curated interaction pairs coming from small-scale studies in which multiple lines of evidence support each interaction[14], a majority of the curated yeast protein pairs were supported by only a single piece of evidence, and came from from experiments that yielded many pairs (Supplementary Fig. 1c, d). It's unclear if and which of these studies applied the quality standards necessary for high-throughput approaches to ensure the release of high-quality datasets, or if they simply produced many hits, of which only a few were followed up in the published study and the rest left unvalidated.

To comprehensively and experimentally assess current interactome information and reduce biases, we developed a yeast two-hybrid (Y2H) version, Y2H-v4, engineered to exhibit orthogonal detection characteristics comparable to previous Y2H versions[11]. This was achieved by using higher copy-number vectors to elevate expression levels, and a longer linker to the activation domain to reduce steric constraints (Supplementary Fig. 1e). We benchmarked this assay against a widely used Y2H version, Y2H-v1[15], along with the *Gaussia princeps* protein complementation assay (GPCA)[16], and the mammalian protein–protein interaction trap (MAPPIT)[17] assay using second-generation positive and random reference sets for *S. cerevisiae* (scPRS-v2/scRRS-v2). Y2H-v4 yielded an assay sensitivity of 22%, with none of the negative control pairs scored positive (Fig. 1d, Supplementary Fig. 1f, and Supplementary Data 2–4). This is a substantial improvement over Y2H-v1, with 24% of detected pairs undetected by any of the other three assays, demonstrating a substantial level of orthogonality for Y2H-v4 (Fig. 1e) and is therefore well suited to assess the quality of PPI datasets with minimal risk of circularity and bias.

We then used these three assays to systematically evaluate the quality of major yeast interaction datasets representing the full range of mapping approaches, by testing representative samples and comparing the recovery rates against scPRS-v2 and scRRS-v2 (Fig. 1f–h, Supplementary Figs. 1g–k, 2a–c, 3, Supplementary Note 1, Supplementary Tables 1–3, and Supplementary Data 5–12). Specifically, we tested 8999 pairs in Y2H-v4, which included all protein pairs with experimental 3D structures curated in 2017 (I3D-exp-17), literature pairs with multiple evidence, with at least one method that detects binary PPIs (Lit-BM-17), and three systematically generated binary maps (Y2H-union-08) (Fig. 1f and Supplementary Data 10). Since most of the systematic binary approaches and many of the studies that comprise Lit-BM used the Y2H assay, we also tested Lit-BM-13, Y2H-union-08, and DHFR-PCA in GPCA and MAPPIT (Fig. 1g, h and Supplementary Data 11). Of all datasets tested, we found that I3D-exp-17, Lit-BM-13/–17, and Y2H-union-08 tested comparably to the positive control set, whereas the other datasets, which make up the majority of the published pairs, tested at rates closer to the random pairs.

Taken together, our rigorous experimental evaluation of existing yeast interactome datasets shows that only a few contain predominantly direct contacts between proteins. Importantly, interactions in I3D-exp-24 and Lit-BM-24 are concentrated around few highly-studied proteins. In contrast, Y2H-union-08 covers the search space more uniformly, although some of the proteins that were missing from previously used clone collections also have fewer interactions here (Fig. 1i and Supplementary Data 13). We then combined I3D-exp-24, Lit-

BM-24, and Y2H-union-08 into a high-quality dataset representing validated binary yeast interactome data in 2024 (ValBin-24), containing a total of 9817 PPIs (Supplementary Data 14). Using four different estimates of the size of the yeast interactome[10–13], ValBin-24 covers between 25 and 50% of the complete PPI network (Fig. 1j and Supplementary Fig. 4a).

In summary, only a small proportion of interactions in public repositories can be assigned to datasets that pass benchmarked experimental validation for binary interactions. Interactions in these datasets cover only a small proportion of the estimated direct interactome in yeast, and so the majority of interactions remain to be discovered by either new experimental or computational approaches. Having established a validated combined dataset of existing PPIs, it is now possible to compare computational and experimental interactome mapping approaches using yeast as a testbed. Notably, of all the experimental methods, systematic high-throughput Y2H-based mapping has proven to be a particularly practical and scalable approach, having produced multiple large interactome maps of reliable and primarily direct PPIs.

## An experimental reference map adds many novel PPIs

Given the limitations of current maps, we first aimed to investigate how much an additional experimental effort can add to our current knowledge of the yeast interactome. We used Y2H-v4 as the primary screening assay, as its orthogonal detection profile maximizes the potential for discovery (Fig. 1d, e, Supplementary Fig. 1f, and Supplementary Data 4). A limitation of previous mapping efforts[11,18,19] was the incompleteness of available ORFeome collections, each covering only 69–77% of the search space of pairwise protein combinations. To generate a more complete map, we compiled a high-quality ORFeome collection covering 99.5% of the 5883 yeast protein-coding genes by starting with the existing FLEXGene collection covering 4933 ORFs[20], and cloning an additional 921 ORFs, thus generating a collection of 5854 sequence-validated ORFs (Fig. 2a and Supplementary Data 15). This ORFeome can be readily employed in different assays[21] and thus forms a foundation for the systematic completion of the *S. cerevisiae* interactome map using assays with complementary detection profiles[22].

We performed three independent screens of the full search space using Y2H-v4, followed by verification of all detected pairs in two independent pairwise tests. Only pairs that were sequence-confirmed and scored positive in both verification tests were included. Thus, we generated a high-quality map of 1880 heteromeric PPIs between 1346 proteins, and 30 homodimeric PPIs, which we refer to as the "yeast reference interactome" (YeRI) dataset (Supplementary Data 16). Dataset quality was then assessed by testing every pair, alongside the scPRS-v2/scRRS-v2 reference sets, in the orthogonal MAPPIT[17] and GPCA[16] assays: in both assays, the pairs in our dataset were recovered at rates similar to Lit-BM-13 (Fig. 2a, Supplementary Fig. 4b, and Supplementary Data 11) demonstrating that the biophysical quality of the generated dataset is on par with high-quality interactions from the literature. We found significant enrichment for interactions between proteins that share annotations for cellular compartments, pathways, or protein complexes (Supplementary Fig. 4c). For a deeper analysis, we made use of profile similarity networks (PSNs) from three systematic and complementary genome-wide functional datasets: (i) genetic interaction similarity[23]; (ii) similarity of growth phenotypes of single-gene knock-out strains in multiple conditions[24]; and (iii) co-expression correlations[25] (Fig. 2b). PPI pairs in our dataset were significantly enriched in each of the three PSNs (Fig. 2c), with higher enrichment among more strongly correlated pairs (Supplementary Fig. 4d).

Interactome networks have a modular organization, knowledge of which facilitates systems understanding and functional annotation of unknown genes[26]. To find functional network modules, higher density maps are advantageous[27], while systematic maps

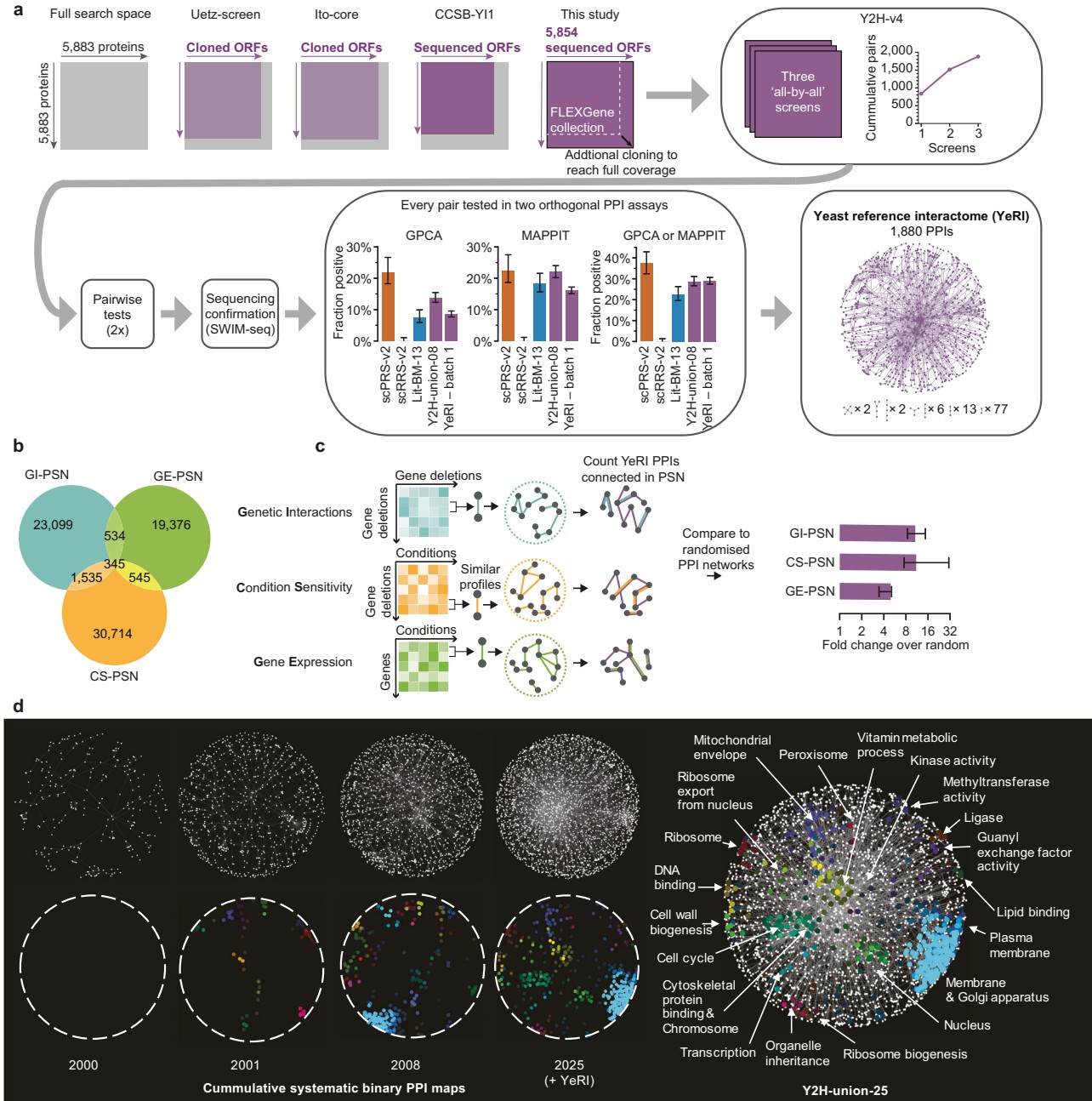

**Fig. 2 | An experimental reference map adds a large number of novel PPIs.**
**a** Generation of the yeast reference interactome (YeRI), including a comparison of our expanded ORFeome collection to previous systematic Y2H maps, screening saturation, and the results of the experimental validation in GPCA and MAPPIT. Error bars are 68.3% Bayesian credible intervals. *n* = 108 (scPRS-v2), 198 (scRRS-v2), 200 (Lit-BM-13), 567 (Y2H-union-08), 994 (YeRI, batch 1, GPCA), 1481 (YeRI, batch 1, MAPPIT) tested pairs. **b** Overlap of connected pairs in the three systematic functional profile similarity networks (PSNs), formed from the highest 1% PCC values for each network. **c** Enrichment of YeRI for interacting protein pairs whose genes are directly connected in functional PSNs, relative to degree-preserved random networks. Central values are relative to the mean, and error bars relative to the inner 68.3%, of the random networks. **d** Enriched clusters of shared functional annotations in Y2H-union-25, over-time, with the addition of each dataset.

avoid the confounding effects of study bias[11]. Therefore, we combined YeRI with the three previous systematic Y2H maps, Uetz-screen[18], Ito-core[19], and CCSB-YI1[11], resulting in Y2H-union-25 comprising 4307 PPIs (Supplementary Data 17). We found that the number of clusters of interacting proteins enriched for shared gene ontology (GO)[28] increased substantially with the addition of each map, with new clusters including: cell cycle, kinase activity, and transcription (Fig. 2d and Supplementary Fig. 4e). Even though *S. cerevisiae* is one of the most extensively-studied organisms, to this day the functions of almost one thousand yeast proteins remain

unknown[29]. Y2H-union-25 contains substantially more interactions involving proteins encoded by genes of unknown function compared to the literature-derived map (Supplementary Fig. 4f–h), highlighting the value of Y2H-union-25 in functionally characterizing proteins. Altogether, 33% of uncharacterized proteins have at least one interaction in Y2H-union-25. To explore whether these interactions could inform function, we applied a guilt-by-association approach to Y2H-union-25 and found that it could successfully predict gene function (Supplementary Note 2, Supplementary Fig. 4i, and Supplementary Data 18).

Notably, the overall sensitivity of YeRI is lower than the 22% scPRS-v2 recovery (Fig. 1d) due to the sampling sensitivity[30] of the screen. Adding YeRI to the union of datasets we previously identified as high-quality binary interactions (ValBin-24) results in a total of 11,349 PPIs. Thus, a substantial fraction of the interactome remains to be discovered. Next, we therefore investigated the potential of computational approaches to achieve completion.

## Assessing an AI-predicted yeast interactome

As an alternative to experiments, computational prediction of PPIs has long offered the promise of a cheaper and faster route to complete contactome maps. Recently, success in protein structure prediction[5] has raised the hope that using the same algorithms to predict interacting proteins could map the contactome at a fraction of the cost of experimental approaches. This computational mapping has been performed for the yeast contactome using AlphaFold/RoseTTAFold (AF/RF)[2]. To assess the accuracy of these predictions, we experimentally evaluated the yeast AF/RF dataset, performing Y2H-v4 pairwise tests of all 1505 pairs. When binning protein pairs according to their "contact probability"−the confidence score assigned to each predicted structure−recovery rates in the experimental assay improve substantially with increasing contact probability. Specifically, pairs with a contact probability of 0.95 or higher tested positive at rates statistically indistinguishable to scPRS-v2, thus indicating high data quality ($P = 0.09$, one-sided Fisher's exact test, Fig. 3a and Supplementary Data 10, 19). We thus restricted subsequent analyses to the 969 out of the 1505 AF/RF pairs (64%) with contact probability > 0.95, which we refer to as "AF/RF-core" (Fig. 3b). Of these, 325 PPIs (34%) had no previous experimental structure available.

Visualizing the network, we realized that the AF/RF-core formed a highly fragmented network (Fig. 3c). Exploring if this could be due to the smaller size of the map, we used random edge removal simulations to compare this to I3D-exp-24, Lit-BM-24, and Y2H-union-08. This analysis revealed a similarity of the two structural networks and showed that it is not the smaller size giving rise to the different topology (Fig. 3d). Upon closer inspection, we noticed that both structural networks lack highly connected hub proteins (Fig. 3e and Supplementary Fig. 5a), which are important for interconnectivity. The underlying reason is likely that hubs commonly interact with many partners via intrinsically disordered regions (IDRs)[31], which pose a challenge for structural approaches, and so might be more amenable to different computational approaches[32]. Additionally, experimental PPI structures are biased towards stronger interactions, potentially because transient interaction pairs are more difficult to crystallize[33]. Computationally predicting such PPI structures could offer a way to overcome this bias in experimental data generation. However, as previously observed[2], AlphaFold also generates more confident predictions for PPIs with larger interfaces (Fig. 3f and Supplementary Data 20). Together, these observations suggest that AF/RF-core not only recapitulates the bias towards more stable PPIs in its training data but actually increases this bias through the PPIs for which it can generate confident structure predictions. It also recapitulates the bias towards better-studied proteins from its training data, as shown by the search space coverage ranked according to the number of publications per gene (Fig. 3g).

Finally, we assessed the novelty of the predicted interaction pairs in the AF/RF-core dataset based on their presence or absence in pre-existing datasets (Fig. 3h). Of the 969 AF/RF-core pairs, 65% correspond to pre-existing experimental structures published elsewhere with either the exact same yeast protein pairs (407 pairs) or homologous protein pairs, referred to as "interolog structures"[34] (225 pairs). Of the remaining 337 AF/RF-core pairs, 167 had previously been reported in high-quality binary datasets. Thus, of 969 high-quality pairs in AF/RF-core, 799 (82%) had been identified previously. Among the 170 remaining pairs, 140 reside in previously described co-complex association datasets. Even though these are not strictly novel interactions, the protein pairs are informative since AF/RF-core upgraded them from co-complex association status to genuinely contacting interaction pairs. Finally, 30 pairs (3%) of the AF/RF-core dataset can be considered strictly novel PPIs that have not been described in any dataset so far.

This surprising finding raised the question: does such a low rate of true novelty also apply to our experimental dataset? Therefore, we conducted the same analysis for all interaction pairs in YeRI (Fig. 3h). In contrast to AF/RF-core, 1382 out of 1880 (74%) of YeRI pairs correspond to strictly novel PPIs not observed previously, whereas 26% were previously observed in experimental structures (135 pairs), interolog structures (93 pairs), high quality binary PPI datasets (169 pairs), and co-complex associations (101 pairs). Thus, the experimental YeRI provides 46-fold more strictly novel PPIs than the AI-based AF/RF-core dataset (1382 versus 30 interactions).

To provide a time-controlled side-by-side comparison of how computational and experimental approaches perform in finding novel interactions relative to what was known at the time of their release, we plotted the growth over time of the ValBin-25 dataset, comprised of Lit-BM, I3D-exp, AF/RF-core, and Y2H-union-25 (Fig. 3i and Supplementary Data 21). Beginning with the first PPIs reported in yeast several decades ago, the total had increased to 494 PPIs by 2000, the time the first systematic binary dataset (Uetz-screen)[11,18] was published. Thus, at that time, with its 577 novel pairs, Uetz-screen provided a delta beyond I3D-exp and Lit-BM of 117%, doubling the number of known yeast PPIs. Subsequent systematic binary maps, Ito-core[11,19] in 2001 and CCSB-YI1[11] in 2008, provided a delta of 44% and 35%, respectively. In subsequent years, Lit-BM and I3D-exp continued their linear growth with over 8000 PPI pairs reported by the time AF/RF-core and YeRI were released. In this context, AF/RF-core and YeRI provided a 4% and 15% delta of pairs beyond previously known binary *S. cerevisiae* PPIs, respectively.

## Testing an updated AI human interactome

A recent AF/RF predicted human protein interactome map[3]−with improvements including a deeper multiple sequence alignment, an updated screening algorithm, and multiple optimized selection cutoffs −computationally screened ~200 million protein pairs and predicted 18,316 interaction pairs with high confidence. This has the potential to outperform the previous approach in yeast. To assess the overall success rate of this approach, we tested a random sample of 3222 of the predicted pairs in Y2H-v1 alongside hsPRS-v2, hsRRS-v2, and Lit-BM-24 controls. The predicted dataset was generated by integrating multiple different strategies, each with a corresponding threshold that depended on whether the protein pair came from the de novo screen or had prior supporting information, e.g., from genetic interaction datasets or PPI databases. Analyzing the Y2H results by these strategies found that, analogous to the yeast predictions, the majority of the data was of high-quality (Fig. 4a, Supplementary Fig. 5b, and Supplementary Data 22). Filtering the dataset by increasing the thresholds to obtain recovery rates not significantly lower than Lit-BM-24 in each strategy resulted in an AF/RF-core dataset of 15,799, corresponding to 89% of the predicted PPIs.

Next, we assessed the novelty of the interaction pairs. Similar to yeast, the human pairs (12,930 PPIs/82%) had largely been observed previously in experiments (Fig. 4b). Again, this contrasts to the experimental human reference interactome (HuRI)[15] dataset, where only 8358 PPIs/16% had been previously observed. Notably, the lower-confidence pairs that we excluded from AF/RF-core were disproportionately strictly novel (Fig. 4c). By plotting human literature-curated binary pairs with multiple evidence, curated experimental structures, and systematic binary maps[15,35−38], over time (Fig. 4d), we observe that our three empirically validated systematic maps of the

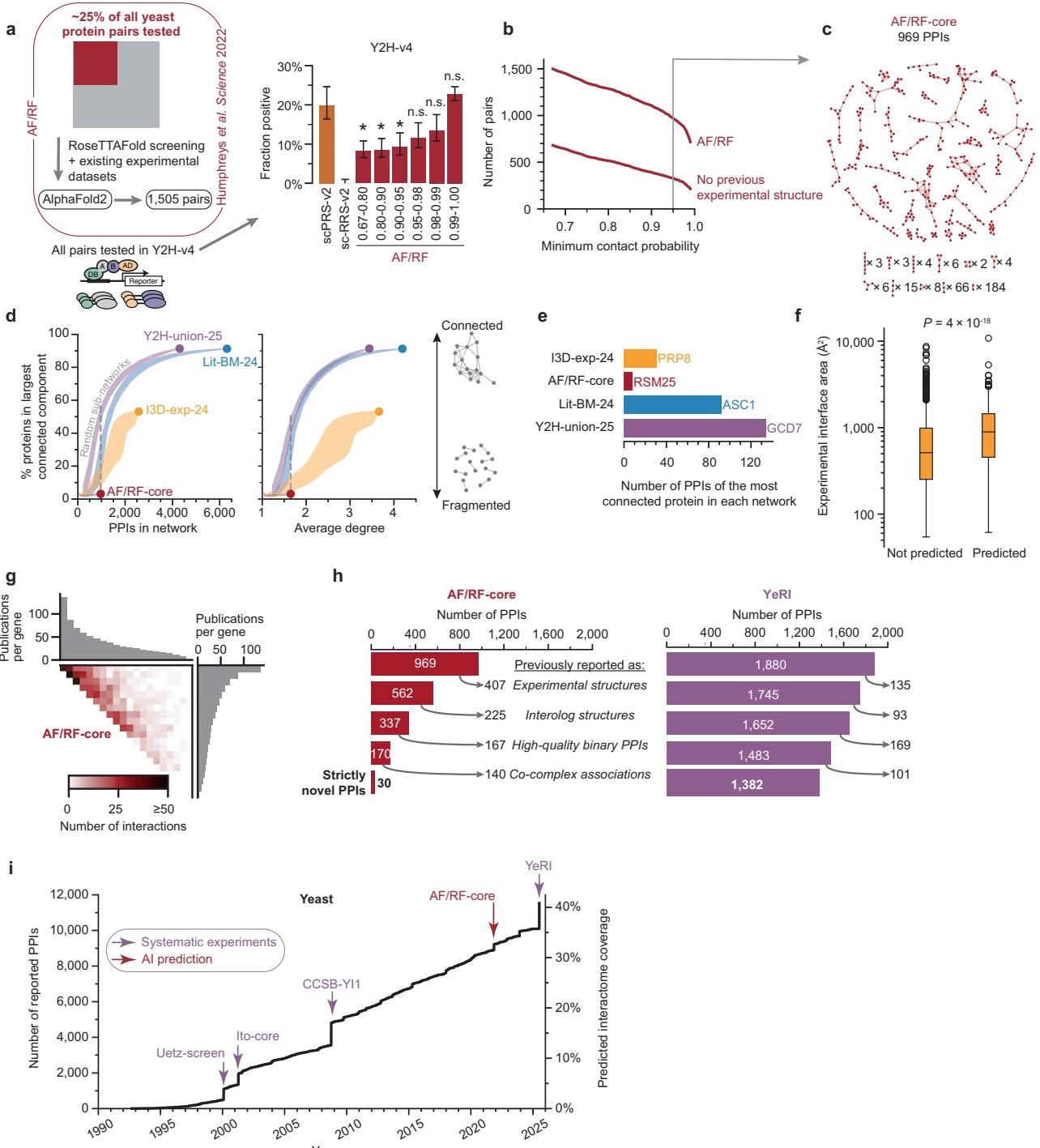

**Fig. 3 | AlphaFold predicts only a small number of novel interactions. a** Results of testing the yeast AF/RF dataset in Y2H-v4, split by contact probability. Error bars are 68.3% Bayesian credible intervals. *$P < 0.05$, n.s.: not significant. One-sided Fisher's exact test, compared to scPRS-v2. From left-to-right: $P = 0.0072$, 0.012, 0.029, 0.087, 0.17, 0.79, $n = 171$, 142, 108, 104, 104, and 567 successfully tested pairs. **b** Size of the AF/RF dataset with increasing contact probability threshold. Number of pairs without a previously existing experimental structure, including interologs, is also shown. **c** Network of the AF/RF-core subset of pairs with contact probability above 0.95. **d** Fraction of proteins in the largest connected component, with circles showing the values for the high-quality contactome maps, and shaded bands showing random subsamples of each network across a range of sizes, plotted against the number of PPIs (left) and the average degree per protein (right). Shaded

bands indicate the innermost 90% interval of random subnetworks. **e** Highest degree proteins in each different high-quality binary interactome map. **f** Distribution of PPI interface sizes, using the experimental structures, split by whether the PPI appears in the predicted structures dataset. Box plot shows median, interquartile range (IQR), and $1.5 \times$ IQR (with outliers). *P*-value calculated using two-tailed Mann–Whitney *U* test, $n = 1395$ (not predicted) and 366 (predicted) PPIs. **g** A heatmap of PPI density of AF/RF-core, within the proteome-by-proteome search space, ordered by the number of associated publications per gene. **h** Yeast AF/RF-core (left) and YeRI (right) categorized based on the presence of each PPI in previous datasets. **i** Cumulative growth of high-quality binary yeast interactome maps over time, with large-scale datasets highlighted.

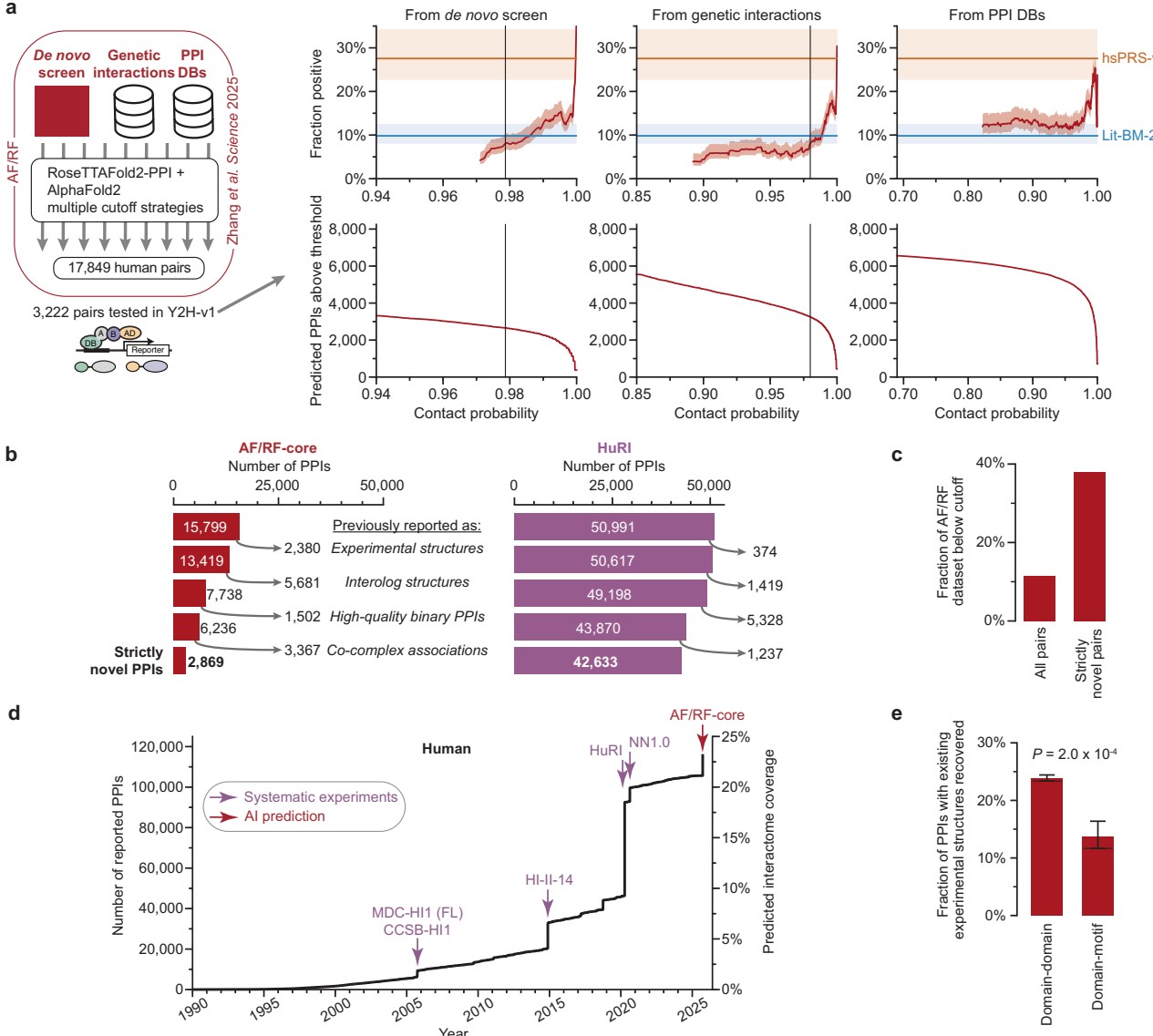

**Fig. 4 | Testing an updated AI-predicted human interactome map. a** Top row: results of testing the human AF/RF dataset in Y2H-v4, as a sliding window across contact probability. Red: AF/RF, blue: Lit-BM-24, orange: hsPRS-v2. Error bands are 68.3% Bayesian credible intervals. Bottom row: number of predicted pairs from corresponding strategy as a function of increasing contact probability. Vertical lines indicate experimentally derived threshold where the mean of the sliding window crosses the lower bound of Lit-BM-24. **b** Human AF/RF-core (left) and HuRI (right) categorized based on the presence of each PPI in previous datasets. **c** Fraction of AF/RF pairs below the experimentally derived thresholds, for all pairs and strictly novel pairs. **d** Cumulative growth of high-quality binary human interactome maps over time, with large-scale datasets highlighted. **e** Fraction of domain-domain and domain−motif mediated benchmark human PPI datasets that are present in AF/RF-core. Error bars are 68.3% Bayesian credible intervals. *P*-value calculated using one-tailed Fisher's exact test (660 domain-domain and 212 domain−motif PPIs).

human interactome (CCSB-HI1[30,35], HI-II-14[37], and HuRI[15]) provided a delta of 39%, 61%, and 100%, respectively, at the time of their release. In contrast, AF/RF-core provided a 10% delta.

Due to the observations that AF/RF-core had lower success with smaller PPI interfaces (Fig. 3f) and missed highly-connected hubs in the yeast network (Fig. 3e), we hypothesized that AI approaches could struggle with domain−motif mediated PPIs, which have smaller interaction interfaces and potentially underlie many hubs in the direct PPI network. We tested this by separating experimental structures into domain-domain and domain−motif subsets, and found that the recovery of domain−motif interactions was less than half of domain-domain PPIs (Fig. 4e and Supplementary Fig. 5c). Since a large proportion of the interactome may be mediated through short linear motifs[39], especially among undiscovered pairs, this could offer a partial explanation for why

AlphaFold-based approaches have had limited success in discovering novel PPIs.

## Integrated contactome mapping strategy

While systematic experimental approaches currently outperform AlphaFold in identifying previously unknown contacting protein pairs, they do not provide structural resolution (Fig. 1a). We wondered for what proportion of novel YeRI pairs could AlphaFold generate a confident structural prediction. To address this, we first optimized and benchmarked algorithmic assay performance using scPRS-v2/scRRS-v2, removing pairs with existing experimental structures, to avoid training bias.

Firstly, we evaluated if AlphaFold3[6] offered a significant improvement over AlphaFold2 for PPI prediction. Surprisingly, we

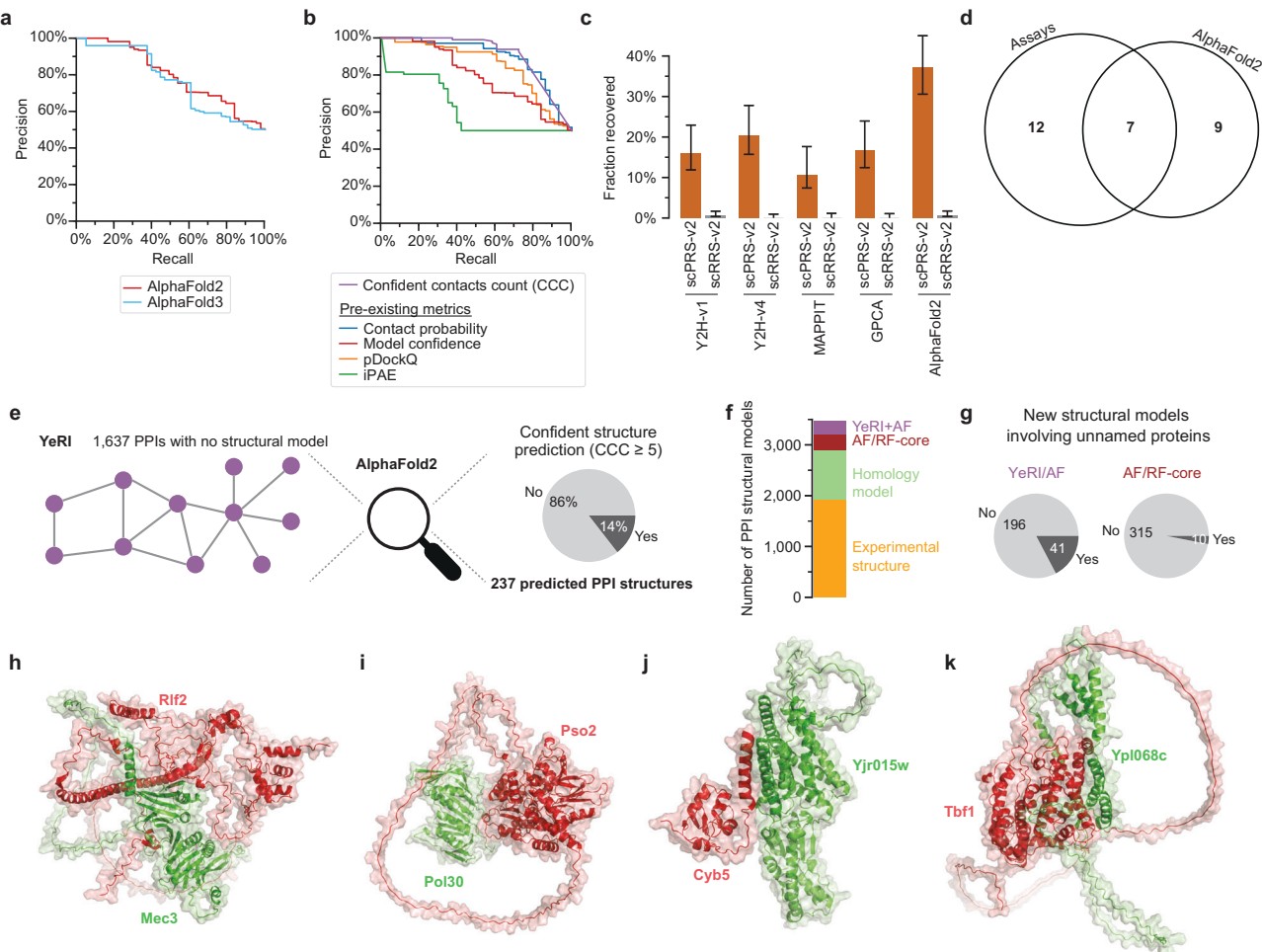

**Fig. 5 | Experimental PPI mapping combined with AlphaFold generates a high-resolution contactome map. a** Precision–recall curve comparing the performance of AlphaFold2 and 3 using positive and random reference sets (scPRS-v2 and scRRS-v2), using the model confidence metric. **b** Precision–recall curve comparing the performance of different confidence metrics for AlphaFold2 on scPRS-v2 and scRRS-v2. Benchmarking four binary PPI assays and AlphaFold2 with the CCC metric, using positive and random reference sets (scPRS-v2 and scRRS-v2), showing recovery rates (**c**) and overlap (**d**). Error bars are 68.3% Bayesian credible intervals. PPIs with existing experimental structures, including homologs, were excluded from these analyses, resulting in $n = 44$ scPRS-v2 pairs and 198 scRRS-v2 pairs. **e** Fraction of YeRI PPIs with no previous structural model for which a confident predicted structure could be generated. **f** Number of PPIs with structural models by source. **g** Fraction of PPIs involving unnamed proteins for new structurally modeled PPIs contributed by YeRI (left) and AF/RF-core (right). **h**–**k** Examples of AlphaFold2 structural models for novel YeRI PPIs.

observed a degradation in performance at low false positive rates due to several highly confident predictions of interactions between random pairs (Fig. 5a and Supplementary Fig. 6a). Therefore, we continued this analysis using AlphaFold2. Secondly, we investigated the performance of different scoring metrics, which aggregate the output of the structural prediction to estimate the likelihood that a pair of proteins interacts. Several scoring metrics have been used in different studies, including model confidence[5], contact probability[2], pDockQ[40], and iPAE[41]. To address a key limitation of existing metrics —their sensitivity to disordered regions outside of the true interaction interface—we also designed and then evaluated the "confident contacts count" (CCC), defined as the number of residue-residue contacts[9] with an associated predicted aligned error (PAE) ≤ 4 Å. Indeed, we find that our CCC outperforms the existing metrics at differentiating between real interactions and random pairs (Fig. 5b and Supplementary Data 23). Using CCC with a cutoff set to match the precision of the experimental assays, i.e., to avoid false positive predictions, we found that confident structures could be generated for 37% of the scPRS-v2 pairs (Fig. 5c and Supplementary Fig. 6b). Interestingly, the detection profile of AlphaFold2 complements the evaluated experimental assays, highlighting the benefits of, and

likely the need for, combining experimental and AI approaches (Fig. 5d).

Following this optimization, we used AlphaFold2 with a CCC cutoff of at least 5 residue-residue contacts with PAE ≤ 4 Å to attempt predicting structures for the novel pairs in YeRI. High-confidence structures could be predicted for 237 of the 1637 PPIs (14%) without previous structural models in YeRI (Fig. 5e, Supplementary Fig. 6c, and Supplementary Data 24). The lower proportion of successful Alpha-Fold2 predictions in the novel YeRI pairs relative to scPRS-v2 could be due to a greater proportion of transient interactions, interactions involving disordered regions, and interactions between less evolutionarily conserved proteins detected in YeRI compared to the reference set[42], in addition we noticed a bias in the scPRS-v2, which contains PPIs with significantly larger interface sizes on average ($P = 1.7 \times 10^{-6}$, two-sided Mann–Whitney $U$ test). While the number of new structural PPI models from YeRI (237) is similar to AF/RF-core (325) (Fig. 5f), the modeled YeRI pairs involve more unnamed genes (Fig. 5g), possibly due to the more uniform coverage of the search space (Supplementary Fig. 6d).

Newly discovered PPIs in YeRI, for which high-confidence structural models could be predicted, offer a rich source of mechanistic

hypotheses. One example involves the chromatin assembly complex subunit Rlf2, known to deactivate the DNA damage checkpoint following DNA repair[43] through an unknown mechanism. The integrated approach reveals a novel structurally modeled interaction between Rlf2 and the DNA damage checkpoint protein Mec3 (Fig. 5h), suggesting a direct regulatory role. Another PPI that could add to our understanding of DNA repair was between the nuclease Pso2 and the sliding replication clamp for DNA polymerase delta Pol30 (Fig. 5i). In addition to illuminating known pathways, several structurally modeled novel YeRI PPIs point to putative functions for uncharacterized proteins. Yjr015w, whose deletion strongly perturbs lipidomics profiles[44], interacts with the sterol biosynthesis protein Cyb5 (cytochrome b5) (Fig. 5j). Similarly, Ypl068c, localized to the nucleus[45] and expressed in response to the DNA-damaging agent MMS[46], interacts with Tbf1, a telomeric repeat-binding factor that is also involved in DNA double-strand break repair[47] (Fig. 5k).

In total, these 237 novel PPIs with high-confidence predicted structures provide a valuable resource of potential mechanistic insight. The combined experimental and computational approach of discovering PPIs using systematic high-throughput experiments followed by AI structural prediction offers a state-of-the-art method for generating a high-coverage and high-resolution map of the contactome; however, further improvements are needed on both sides to generate a completed map of the protein contactome.

## Discussion

We have developed an experimental framework to rigorously assess the quality of current interactome datasets and the contributions by state-of-the-art AI tools. Firstly, we determined that datasets composed of high-quality direct interactions constitute only ~5% of the 200,000 yeast pairs reported in manually curated databases, whereas the vast majority of pairs are either indirect associations in larger complexes or protein pairs that do not stand up to experimental scrutiny. The more than 11,000 high-quality binary contact interactions in ValBin-25 constitute a gold-standard foundation for diverse future applications requiring high-quality direct interaction information. Given that the number of subunits in a complex strongly influences the proportion of indirect associations versus direct interactions, it is probable that additional directly contacting pairs could be retrieved from small protein complexes in AP/MS datasets.

The incredible success of AlphaFold in predicting protein tertiary structures[1] has sparked hopes that its application to quaternary structure prediction for protein pairs could be an effective approach to completing the interactome. Indeed, we find that the confidently predicted pairs pass experimental scrutiny and have a quality that matches experimental data, although this requires a more stringent cut-off than initially suggested[2]. At the same time, it is clear that, with regard to interaction discovery, that current AlphaFold versions do not provide a quantum leap of progress, but contribute relatively modest numbers of novel PPIs. This is in stark contrast to the experimental approach, which contributed 46-fold more novel PPIs. The predicted PPIs are biased towards interactions with larger interfaces, as observed before[2], whereas interactions with short-linear motifs tend to be underrepresented. An important consequence of the latter observation is that the predicted networks, similar to structurally resolved interactome networks obtained from experiments, lack central hubs and a highly-connected topology characteristic of protein interaction networks. At this stage, it must be concluded that computational interaction identification by AlphaFold-related tools is not competitive with experimental approaches.

The purpose of AlphaFold is the prediction of structures, and a powerful contribution of this tool is the generation of quaternary structure predictions, thus providing the highest level of contact details for the respective pairs. Intriguingly, AlphaFold is able to predict high-confidence structural models for many pairs that were missed by the virtual screen but have been identified by experimental approaches. For YeRI, AlphaFold could infer new structural models for 237 PPIs, of which 72% (171) were not in the AF/RF search space because the pair had few homologs (shallow MSA depth), whereas 28% (66) were tested and were below the thresholds. These could either have been missed by the screening algorithm whilst being detected by the experimental screening, or could have been recovered due to the improved CCC interaction score for AlphaFold structures, which we developed based on the scPRS-v2/scRRS-v2 benchmarking data, and which outperformed the previously described metrics.

Looking forward, some fraction of the cases where predicted structures for interactions were missed could have been a result of search space limitations related to the computational pipeline. However, even ignoring such technical hurdles that can be overcome with more resources, many protein interaction pairs in YeRI remain for which no structure could be calculated by the algorithm. These point to more fundamental issues for structural interaction modeling that may be related to biochemical differences guiding the formation of quaternary, in contrast to tertiary structures. One contributing reason is likely the scarcity of available experimental structures for heteromers compared to monomers[48]. Furthermore, the diversity of interaction structure types in nature exceeds the diversity of protein folds[49]. Thus, compared to the tertiary folding problem, the quaternary folding problem is more complex, while less data is currently available. Finally, other elements that are known to be critical for PPI formation may not be effectively captured by current structure-prediction methods, such that modifications to the computational strategies may be necessary. Many PPI interfaces involve IDRs[39], posing a significant challenge to structure-based approaches, and so alternative computational approaches may perform better for those PPIs[32]. Thus, for the foreseeable future, experimental and computational approaches will have to complement each other in generating increasingly complete and structurally resolved contactome maps.

Finally, we present a yeast reference interactome map, YeRI, that adds more than 1300 novel interaction pairs to our current knowledge. As we show, this constitutes a powerful resource of biologically relevant protein interactions, especially when combined with other high-quality systematic binary datasets represented in Y2H-union-25. This progress was achieved using an alternative screening assay, by improvements to the screening pipeline, and by a completed ORFeome collection that represents essentially all 6000 genes in the yeast genome. This foundational ORFeome eliminates an important variable, namely search space coverage, from future experimental operations. For subsequent interactome mapping iterations, emphasis can be placed on complementary detection profiles of different interaction assays[22]. Doing so, it can be expected that each assay may yield a similar number of novel PPIs as we identified here to increasingly approach completion of the contactome at the level of protein interactions. We have previously shown that a limited number of different assays and assay versions have a combined sensitivity up to 80%[22]. Reducing the costs associated with the required screening effort will be facilitated by the implementation of improved screening and pooling strategies[50–53]. Along this line, we anticipate parallel improvements by the AI community on the identification and, more importantly, on the structural modeling of quaternary structures of all known interaction pairs to jointly complete a structurally resolved contactome map.

## Methods
### Yeast strains
Yeast haploid strains *MAT*α Y8930 and *MAT***a** Y8800, derived from PJ69-4[54] and used previously[11,55] harbor the following genotype: *leu2-3,112 trp1-901 his3Δ200 ura3-52 gal4Δ gal80Δ GAL2::ADE2 GAL1::HIS3@LYS2 GAL7::lacZ@MET2 cyh2^R*. Yeast cells, parental strains or

transformants, were cultured either in YEPD or synthetic drop out media, supplemented as needed and incubated at 30 °C.

## Bacterial strains

Chemically competent DH5α or DB3.1 *E. coli* cells were used for all bacterial transformations in this study. Transformed cells were cultured in Luria broth or Terrific broth, supplemented with antibiotics (50 μg/ml of ampicillin, spectinomycin, or kanamycin) as needed and incubated at 37 °C.

## Human cell lines

Human embryonic kidney HEK293T cells were cultured in Dulbecco's Modified Eagle Medium (DMEM) supplemented with 10% fetal bovine serum, 2 mM L-glutamine, 100 I.U./ml penicillin, and 100 μg/ml streptomycin. Cells were incubated at 37 °C with 5% $CO_2$ and 95% humidity.

## Yeast open reading frames

The list of yeast ORFs was downloaded from the Saccharomyces genome database (SGD) (https://www.yeastgenome.org/) on January 14th, 2017. Four ORFs (YCR097W/HMRA1, YCR096C/HMRA2, YCL066W/HMLALPHA1, YCL067C/HMLALPHA2) annotated in SGD as "silenced gene" were removed. Only SGD-annotated "verified" and "uncharacterized" ORFs were included, whereas ORFs annotated as "dubious" were excluded, leaving a total of 5883 ORFs with 5155 and 728 ORFs classified as verified and uncharacterized, respectively. All datasets analyzed have been restricted to these 5883 ORFs, and previous ORF names that appear as aliases for one of these ORFs have been mapped to their corresponding new names.

## Assembly and description of yeast biological network datasets

**Y2H-union-08: Uetz-screen, Ito-core, and CCSB-YI1.** As described previously[11], Uetz-screen is a subset of PPIs from Uetz et al.[18,19] that was obtained from a proteome-scale systematic Y2H screen, excluding a smaller-scale, targeted experiment with a smaller number of well-studied bait proteins. Ito-core is a subset of PPIs found three times or more in Ito et al.[19], excluding unreliable pairs of proteins found only once or twice. CCSB-YI1 is a proteome-scale dataset of Y2H PPIs validated using two orthogonal assays, MAPPIT and yPCA[11]. After restricting to PPIs involving the 5883 ORFs (described above), the PPI dataset sizes are as follows: Uetz-screen: 645 PPIs; Ito-core: 816 PPIs; CCSB-YI: 1772 PPIs. The union of these maps (Y2H-union-08) contains 1933 proteins and 2833 PPIs.

## Literature-curated biophysical datasets (Lit-CC, Lit-BS, Lit-BM).

Literature-curated pairs were obtained from the databases MINT[56], IntAct[57], DIP[58], and BioGRID[59]. The data files used were the 2024-11-29 release from IntAct (containing data from IntAct, MINT, and DIP) and BioGRID release 4.4.240 (from 2024-11-25). We excluded evidence corresponding to the systematic, proteome-scale binary and co-complex association datasets[11,18,19,60–64]. Data were filtered to ensure valid IDs for UniProt accession numbers, PubMed IDs, and PSI-MI terms. Each piece of evidence for a protein pair had to consist of a PubMed ID and an interaction detection method term in the PSI-MI controlled vocabulary (http://www.psidev.info/groups/molecular-interactions). Duplicated evidence can arise in cases where different source databases curate the same paper. We merged duplicated entries for each pair, which we identified as multiple pieces of evidence with the same PubMed ID and experimental interaction detection codes, which are either identical or have an ancestor-descendent relationship in the MI ontology. In the latter case, the more specific descendent term was assigned to the merged evidence. In order to select the subset of protein pairs corresponding to binary interactions (as opposed to co-complex associations), we developed a manual classification of the PSI-MI interaction detection method

terms[37]. Our classification has since been updated to cover new experimental methods, which have been added to the controlled vocabulary in the intervening time. The methods are classified into three categories; "invalid", "binary" and "co-complex". Where "invalid" corresponds to PSI-MI terms that are not considered valid experimental PPI detection methods, "binary" corresponds to terms that detect binary PPIs and "co-complex" corresponds to terms that detect potentially indirect associations. An example term in the "invalid" category is "colocalization". All protein pairs annotated with "invalid" terms were excluded. "Binary" versus "co-complex" evidence was used to categorize protein pairs in the literature-curated dataset as follows. Pairs with no binary experimental evidence were classified as "Lit-CC", corresponding to 158,843 pairs. Pairs with a single piece of binary evidence and no other evidence were classified as "Lit-BS", corresponding to 15,090 pairs. Finally, pairs with two or more pieces of evidence, including at least one binary evidence, were classified as "Lit-BM", corresponding to 6309 pairs.

Previous literature-curated datasets generated in 2017 and 2013 were used as a source dataset for pairs experimentally tested in GPCA, MAPPIT and Y2H-v4 (see Engineering of Y2H destination vectors) experiments. These were generated and processed as above, with small differences. Lit-BM-17 and Lit-BS-17 were obtained via the Mentha resource using the data file dated August 28th 2017[65]. Lit-BM-13/Lit-BS-13/Lit-CC-13 were generated as described previously[37]. Yeast PPIs annotated through December 2013 from six source databases: BIND[66], BioGRID[59], DIP[58], MINT[56], IntAct[67] and PDB[68] were extracted and processed using the same protocol.

One note: we excluded the evidence from the systematic, proteome-scale binary and co-complex association datasets based on the PubMed ID, but not all pairs associated to those PubMed IDs are included in our definition of the systematic datasets. For example, for Ito et al., we just take the Ito-core subset. As a result, the bars in Fig. 1c are a few percent short of the total number of pairs reported in literature-curated databases.

## Direct PPIs with experimental structures.

The most definitive proof that two interacting proteins are in physical direct contact is the availability of a three-dimensional (3D) structure of their interface. We used the subset of Interactome3D[9] restricted to experimental structures, excluding homology models. We re-ran the Interactome3D pipeline specifically on *S. cerevisiae* experimental structures, ignoring homology models (Supplementary Data 1) to generate the dataset referred to as "I3D-exp-24", used for most computational analyses. The dataset from the May 2020 release of Interactome3D, referred to as "I3D-exp-20", was used for some computational analyses, specifically to define the homology models of PPI structures, and to define direct vs. indirect contacts within complexes (see the section "direct or indirect contact in a complex structure"). The dataset from the June 2017 release, "I3D-exp-17", was experimentally tested in its entirety using Y2H-v4 (see "engineering of Y2H destination vectors"). The date assigned to PPIs was obtained from the PDB database, taking their earliest release date for all PDB structures from the "complete" Interactome3D dataset.

## Note on the overlap between I3D-exp-24 and Lit-BM-24 PPIs.

There were a surprisingly large number of pairs in I3D-exp-24 that were absent from Lit-BM-24 (1414 pairs in I3D-exp-24 that are not in Lit-BM-24, 1162 pairs in the intersection). Other than those due to the difference between the exact date of generation of the two datasets, the pairs were mostly cryo-EM structures of larger complexes. The reason for this is that in the generation of the literature-curated datasets (see section "literature-curated biophysical datasets"), we didn't use the structural data for direct contacts. Instead, we base the binary vs. co-complex distinction on the experimental method used, and we classify

cryo-EM as co-complex since the information of whether the reported pairs are in direct contact or not is not contained in the data files from the PPI repositories.

**AF/RF predicted structures.** The list of PPIs for AF/RF came from the Excel file captioned "descriptions of all predicted PPIs"[2]. Six PPIs with missing gene names were discarded. The predicted structures, which were available only for pairs with contact probability ≥ 0.9, were downloaded from https://modelarchive.org/doi/10.5452/ma-bak-cepc.

**Systematic AP/MS.** AP/MS-06 is made up of Gavin et al.[61], Gavin et al.[62], Krogan et al.[63]. We didn't include Ho et al.[64], since it was generated with a smaller, more focussed selection of bait proteins and thus was not considered proteome-scale.

**Functional profile similarity networks (PSNs).** Genetic interaction similarity profile data (GI-PSN) were extracted from Costanzo et al.[23]. The average PCC of a pair was used if multiple PCCs were available. Pairs with PCCs ranked in the top 1% were used to generate the GI-PSN. Condition-sensitivity data (CS-PSN) was extracted from Hillenmeyer et al.[24]. The log of growth ratios from the homozygous deletion data were used to calculate PCC for each pair of genes. Pairs with PCCs ranked in the top 1% were used to generate the CS-PSN. Gene expression data (GE-PSN) was downloaded from https://coxpresdb.jp[25]. PCCs were extracted from the union datasets (Sce-m.c3-0 and Sce-r.c1-0, 2018.11.07), and pairs with PCCs ranked in the top 1% were used to generate the GE-PSN. See Supplementary Table 4 for details of the network size and thresholds.

### Generation of scPRS-v2 and scRRS-v2

Due to the changes in yeast ORFeome version used here, we updated our positive reference set (PRS) and random reference set (RRS) from our original set[11]. We named the updated *Saccharomyces cerevisiae* positive and random reference sets "scPRS-v2" and "scRRS-v2", respectively. In Yu et al., 188 PPIs with five or more papers describing each were finalized as PRS candidates, for which 116 protein pairs corresponded to cases where both ORFs were available in the ORFeome collection at the time. Of the 188 PPIs, we filtered for pairs that were also present in Lit-BM-20, and for which both ORFs are available the FLEXGene collection[20], resulting in a final scPRS-v2 of 108 PPIs. Of 188 RRS pairs in Yu et al., we removed all ORFs annotated as dubious, then required they have both ORFs in the FLEXGene collection. We then increased the size of the RRS by adding additional pairs randomly selected from the space of all possible pairwise combinations of ORFs in the FLEXGene collection. Since the RRS is used as a negative control, we then filtered out any pairs that appeared in any of the experimental PPI or co-complex association datasets, which resulted in removing one pair that appeared in Lit-CC-20, resulting in a final scRRS-v2 of 198 pairs.

### Assembly and description of human biological network datasets

**AF/RF predictions for human.** Originally, data corresponding to the Zhang et al. biorxiv preprint v1[69] downloaded from http://prodata.swmed.edu/humanPPI/bulk_download/GoodPairs.txt on 2024-10-05. The information about the GI and PPI-DB sources was obtained from personal communication with the authors (Qian Cong and Jing Zhang). The strategies were recreated using the information in the methods section and validated by comparing to the numbers in Fig. 3A of the preprint, finding only small disagreements of 1–3% with the values. These were used to generate the random sample of pairs to test in Y2H. These were subsequently filtered for pairs in the published Zhang et al. Science[3] version of the dataset, which was also used for all computational analyses.

**Lit-BM-24.** Data were generated as for yeast, above.

### Engineering of Y2H destination vectors

Gateway-compatible 2 μ high-copy destination vectors pVV212 and pVV213[70] with a Gal4 DNA binding domain (DB) and a Gal4 activation domain (AD), respectively, were modified to be compatible with our standard Y2H vectors pDEST-DB and pDEST-AD-*CYH2*[55] with respect to the *LEU2* and *TRP1* as selectable markers. The resulting destination vectors pDEST-DB-QZ212 and pDEST-AD-QZ213 also carry *CAN1* or *CYH2* genes as counter-selectable markers, respectively. The *CYH2* and *CAN1* counterselectable markers facilitate plasmid shuffling for the identification of auto-activators[71]. Gateway LR reactions between yeast ORFs flanked by attL1 and attL2 sites with the attR1 and attR2 sites of pDEST-DB-QZ212 and pDEST-AD-QZ213 result in attB1 and attB2 sites flanking yeast ORFs now fused downstream of either the Gal4 DB or Gal4 AD sequences of the respective destination vector. See Supplementary Tables 5 and 6 for detailed information.

### Benchmarking yeast two-hybrid (Y2H) assay versions

Assay versions were benchmarked using scPRS-v2 and scRRS-v2. The Y2H version with destination clones in vectors pDEST-DB-QZ212 and pDEST-AD-QZ213 was named Y2H version 4 (Y2H-v4), as Y2H-v1, -v2, and -v3 have been described previously[15]. The performance of Y2H-v4 was compared to Y2H-v1, which consists of destination clones in pDEST-AD-*CYH2* and pDEST-DB, and was used to generate CCSB-YI1[11].

The Y2H assay was performed as described previously[37,55]. Briefly, Y8930:pDEST-DB-QZ212-ORF and Y8800:pDEST-AD-QZ213-ORF haploid strains were inoculated and mated in YEPD. After enrichment for diploids in synthetic complete media lacking leucine and tryptophan (SC-Leu-Trp), diploids were spotted on SC-Leu-Trp-His solid media containing 1 mM 3-amino-1,2,4-triazole (3AT), testing for *GAL1::HIS3* activation and on a set of SC-Leu-His+ 1 mM 3AT plates supplemented with 10 mg/l cycloheximide (CHX) to identify spontaneous DB-ORF auto-activators[55]. After 3 days incubation at 30 °C, yeast strains growing on SC-Leu-Trp-His + 3AT solid media and not on SC-Leu-His +3AT + CHX media were scored as positives. The interacting pairs were identified based on plate position.

### Generation of an expanded yeast ORFeome collection

Our version of the yeast FLEXGene clone collection[20] of full-length ORFs cloned in either pDONR201 or pDONR221, both Kan^R, contains 5296 unique ORFs. After comparing with the SGD annotation (version 2014) (https://www.yeastgenome.org/) we removed 26 obsolete ORFs, 18 ORFs with sequences that did not match the reference, 175 dubious ORFs, 126 intron-containing ORFs, and 18 other ORFs no longer annotated as protein-coding. This resulted in 4933 SGD-annotated ORFs. For the 950 SGD-annotated ORFs not present in the yeast FLEXGene collections, entry clones were generated in-house and are referred to as the "supplemental ORFeome collection". ORF sequences were amplified without their native stop codon from either *S. cerevisiae* S288C genomic DNA (ORFs without introns) or cDNA (for ORFs in genes containing introns) using KOD high fidelity polymerase (Novagen) and 18–20 nucleotide ORF-specific forward and reverse PCR primers tailed with Gateway attB1 and attB2 sequences

attB1 forward primer tail:
5′GGGGACAAGTTTGTACAAAAAAGCAGGCTCCACC
attB2 reverse primer tail:
5′GGGGACCACTTTGTACAAGAAAGCTGGGTCCTA

from Hu et al.[20], essentially as described[72]. The CTA sequence in the Gateway tail of the reverse primer provided a synthetic stop codon for all ORFs. Amplified ORFs were transferred to pDONR223 (Spec^R) by Gateway BP recombination cloning (Invitrogen) and transformed into chemically competent DH5α *E. coli* cells. Sanger sequencing of PCR products, generated with universal forward and reverse primers, was used to confirm the identity of all cloned ORFs as described[72]. Nine hundred twenty-one ORFs were obtained using this approach.

The cloned ORFs were longer on average than the previously cloned ORFs in the FLEXGene collection (mean of 1916 nucleotides compared to 1359 nucleotides in FLEXGene, $P = 9 \times 10^{-9}$, two-sided Mann–Whitney $U$ test).

## ORFeome cloning in Y2H destination vectors

To generate an arrayed library of DB-X and AD-Y expressing strains, the yeast ORFs were transferred into both destination vectors, pDEST-DB-QZ212 and pDEST-AD-QZ213, by Gateway LR recombination cloning (Invitrogen). Gateway LR reaction products were transformed into DH5α *E. coli* cells, plasmid DNA was extracted and used to transform yeast strains. pDEST-DB-QZ212 and pDEST-AD-QZ213 expression clones were transformed into yeast strains *MAT*α Y8930 and *MAT***a** Y8800, respectively[55]. To confirm the quality of the collection, we verified the identity of a sample of clones from each plate. We sequenced one row (row E) from each plate of transformed yeast cells using SWIM-seq (see Yeast colony sequencing below) and obtained 91% and 90% sequence confirmation rates for yeast DB and AD clones, respectively.

## Auto-activator detection for filtering before Y2H screening

We tested for auto-activation of the *GAL1::HIS3* reporter gene by AD-ORF or DB-ORF fusion proteins in both haploid and diploid yeast cells. To identify auto-activator clones in haploid yeast cells, Y8930:DB-ORF and Y8800:AD-ORF strains were grown to saturation in SC medium lacking leucine (SC-Leu) or tryptophan (SC-Trp), respectively. After 24 h of incubation, Y8930:DB-ORF and Y8800:AD-ORF haploids were spotted on SC-Leu-His + 1 mM 3AT or SC-Trp-His + 1 mM 3AT to test for *GAL1::HIS3* activation. Viability of the haploids was confirmed by growth on SC-Leu or SC-Trp media, respectively.

To identify auto-activators in diploid yeast cells, *MAT*α Y8930:DB-ORF and *MAT***a** Y8800:AD-ORF strains were mated against their respective opposite mating type strains carrying the corresponding destination vectors without any fused ORFs. Mating was conducted in rich medium, YEPD, and resulting diploids were enriched following growth in SC-Leu-Trp. Diploids were spotted on SC-Leu-Trp-His + 3AT, to test for *GAL1::HIS3* activation, and on SC-Leu-Trp to confirm the viability of the diploids. For both haploids and diploids, after incubation at 30 °C for 3–4 days, strains growing in the absence of histidine and presence of 3AT were considered auto-activators. Altogether 560 DB-ORFs and 1 AD-ORF auto-activators were removed from the final screening collection.

The remaining DB-ORF and AD-ORF clones were re-arrayed into four different groups to separate ORFs with similar nucleotide sequences, defined as BLAST scores of 100 and above. Separation of similar ORFs makes the downstream sequence identification of the short NGS reads more accurate, as the reads are aligned to specific groups of ORFs without sequence ambiguity. Filtering for pairs that passed auto-activator screening and successful cloning resulted in a final collection, which was then used for systematic screening, encompassing 4778 DB-ORF clones and 5700 AD-ORF clones, covering a total of 5854 yeast ORFs.

## Primary yeast two-hybrid (Y2H) screening

Three replicate Y2H screens were performed. Individual *MAT*α Y8930:DB-ORFs were mated in YEPD against a pool of ~700 (FLEXGene collection) or ~200 (supplemental collection) *MAT***a** Y8800:AD ORFs. AD-ORF pool size was decreased for the supplemental collection to facilitate screening. After enrichment in SC-Leu-Trp, 5 μl of the culture was spotted on SC-Leu-Trp-His + 1 mM 3AT solid media and on SC-Leu-His + 1 mM 3AT + 10 mg/l CHX to identify spontaneous DB-ORF auto-activators[55]. After incubation at 30 °C for 3 days, strains growing on SC-Leu-Trp-His + 3AT but not on SC-Leu-His + 3AT + CHX were picked and grown in liquid SC-Leu-Trp. As we used libraries of pools of *MAT***a** Y8800:AD-ORF, it is possible to obtain more than one interaction per

mini-library. To account for that, we picked up to three colonies per growth spot. Cell lysates were prepared from the saturated cultures and used as templates in PCR reactions to amplify and identify the bait and prey sequences[55].

## Yeast colony sequencing

To efficiently and cost-effectively identify both bait and prey proteins for thousands of positive colonies, we used a method called SWIM-seq (Shared-Well Interaction Mapping by sequencing) as previously described[15]. Briefly, DB- and AD-ORFs were simultaneously amplified from 3 μl yeast lysate, using well-specific primers. PCR reactions were performed using Platinum Taq (Life Technologies). After PCR amplification, barcoded PCR products from an entire 96-well plate were pooled together and purified (Qiagen, PCR Purification Kit). The pooled amplicons from each plate were subjected to Nextera "tagmentation" using Tn5 transposase to generate a library of amplicons with random breaks to which the adapters were ligated[73]. We then re-amplified those fragments to generate a library of amplicons such that one end of each amplicon bears the well-specific tag and the other "ladder" end bears the Nextera adapter. A final Illumina sequencing library was prepared by adding plate indexes using the i5 and i7 Illumina adapter sequences. Next-generation sequencing was performed with Illumina Solexa technology, allowing for the identification of interacting first pass pairs of proteins (FiPPs) (see "sequence identification of interacting ORFs"). Due to the small number of pairs to be identified, interacting pairs from the first screen of the supplemental space were amplified with the universal AD and DB forward and reverse primers and ORF sequences were identified by Sanger sequencing (Genewiz). All SWIM-primers (Supplementary Table 7) were synthesized by Thermo Fisher Scientific, whereas the universal AD, DB and term primers were synthesized by Eurofins Genomics.

## Pairwise test

To confirm all FiPPs, pairwise tests were performed in the same DB-X/AD-Y orientation they were found in the primary screens. Briefly, glycerol stocks from Y8930:DB-ORF and Y8800:AD-ORF haploid strains were inoculated in SC-Leu or SC-Trp, respectively. Saturated cultures were mated in YEPD. After enrichment for diploids, yeast cultures were spotted on SC-Leu-Trp-His + 1 mM 3AT solid media, testing for *GAL1::HIS3* activation. Preliminary investigations using four technical replicates demonstrated that in 97% of the cases, the quadruplicates behaved identically (data not shown). Therefore, given the high reproducibility of technical replicates, the culture was spotted only once per selective media. To increase the robustness of our approach, we implemented an additional test to identify de novo auto-activators in which Y8930:DB-ORF strains were mated against a Y8800:AD with no ORF fused to the activation domain (Y8800:AD-Empty ORF) and spotted on SC-Leu-Trp-His + 1 mM 3AT solid media. Diploids that gave rise to growth on SC-Leu-Trp-His + 1 mM 3AT media, but did not grow when the respective Y8930:DB-ORF was mated to Y8800:AD-Empty ORF, were selected as positive interacting pairs of proteins. Positive protein pairs were sequence confirmed, as done for the primary screens as described above. As positive and negative controls, the scPRS-v2 and scRRS-v2 pairs were distributed randomly across the respective mating plates and tested at the same time. For a batch of pairwise testing to be considered successful, we required no more than 1% of RRS and between 10 and 25% of PRS to score positive.

## Sequence identification of interacting ORFs

We used an existing computational pipeline[15] to process demultiplexed paired-end reads returned from Illumina sequencing and identify the interacting ORF pairs from the Y2H screens. Paired-end reads are in fastq format, with one read, R1, containing a part of the ORF sequence and the other paired read, R2, containing the well index.

We used Bowtie 2[16] (v2.2.3) to align all R1 reads to reference sequences and extracted the well-identifying indices from the R2 reads. AD-ORFs and DB-ORFs that shared the same well indices were paired together and called FiPPS. To identify likely true AD/DB pairs, we developed a "SWIM score" $S$[15] that takes into account the AD and DB reads in each well, total reads returned from the sequencing run, and other factors.

$$S = \frac{2}{\frac{a + M}{x} + \frac{d + N}{y}}$$

where $x$ and $y$ are read counts of an AD-ORF and DB-ORF in a given well, respectively, $a$ and $d$ are total read counts of all aligned AD-ORF and DB-ORF in that well, and $M$ and $N$ are pseudo-counts for AD and DB, respectively, which were constant for each sequencing batch but varied for different batches. We then selected FiPPs for pairwise testing using a cutoff that balances the risk of testing too many false positives FiPPs versus not testing too many true positive FiPPs. The cutoff varied for different screens and sequencing runs to adjust for slight variations in the screening and sequencing protocol.

### Validation in orthogonal assays

To assess the precision of various datasets[30], PPIs were validated in two orthogonal assays: mammalian protein-protein interaction trap (MAPPIT)[17] and *Gaussia princeps* luciferase protein complementation assay (GPCA)[16]. As positive and negative controls, we used pairs of scPRS-v2 and scRRS-v2, respectively. For both assays, expression clones were generated by Gateway LR recombination cloning as described above. Expression clones for GPCA were generated by transferring ORFs into pSPICA-N1 and pSPICA-N2 destination vectors[16], each expressing a different fragment of humanized *Gaussia princeps* luciferase (GL1 and GL2)[74]. MAPPIT expression clones were generated by LR transfer of ORFs into pMBU-I-2994 and pMBU-I-4199 destination vectors[17]. After transformation of all expression clones into DH5α *E. coli* cells, plasmid DNA was extracted and purified using Qiagen 96 Turbo kits (Qiagen) on a BioRobot 8000 (Qiagen). Three different GPCA and two different MAPPIT experiments were performed.

***Gaussia princeps* protein-fragment complementation assay (GPCA).** GPCA experiments were performed as described previously[16]. Briefly, on the first day of the assay, ~30,000 to 40,000 HEK293T cells were seeded in each well of a 96-well microtiter plate (Greiner Bio-One). DNA concentration was measured for all clones, and samples were diluted to a final concentration of 25 ng/μl. After a 24 h incubation at 37 °C, confluent cells were transfected with 300 ng of pSPICA-N1-ORF and pSPICA-N2-ORF vectors using polyethylenimine (PEI). After a second 24 h incubation at 37 °C, cells were washed with PBS supplemented with calcium and magnesium chloride. To lyse the cells, 40 μl of 5× diluted Renilla lysis buffer (Promega) were added to each well. The plate was then covered with aluminum foil and agitated at 900 rpm for 30 min at 37 °C for cell lysis. Luciferase activity was measured on a TriStar Berthold Microplate reader by adding 50 μl per well of Renilla luciferase substrate (Renilla Luciferase Assay System, Promega), with a measurement time of 4 s. The measurement score, RLU (relative light unit), was assigned to the tested pair.

**Mammalian protein–protein interaction trap (MAPPIT).** As an additional orthogonal validation assay, MAPPIT experiments were performed as described elsewhere[15,37]. In short, HEK293T cells were grown in 384-well plates and co-transfected with a luciferase reporter and plasmids for both bait and prey fusion proteins. Twenty-four hours post-transfection, cells were either stimulated with ligand (erythropoietin) or left untreated, then incubated for an additional 24 h before luciferase activity was measured in duplicate. The MAPPIT validation experiment was deemed valid if both bait and prey were successfully cloned into expression vectors and bait expression was detected using a luminometer. "Fold-induction" values (signal from stimulated cells divided by signal from unstimulated cells) were calculated for each tested pair, and two negative controls (no bait with prey and bait with no prey). Each tested pair was assigned a quantitative score: the fold-induction value of the pair divided by the maximum of the fold-induction value of the two negative controls.

### Experimental benchmarking of public PPI datasets

PPIs extracted from the biophysical maps described in Supplementary Table 3 were tested in assays Y2H-v4, GPCA, and MAPPIT following the same experimental procedures as described above. A summary of the number of tested pairs in each dataset is available in Supplementary Table 3. Samples, if used, were drawn randomly.

An additional Y2H-v4 experiment was performed to test pairs from the AF/RF dataset, along with scPRS-v2 and scRRS-v2. Roughly half of AF/RF PPIs had already been tested in the first Y2H-v4 experiment, as they overlapped with one or more of the tested datasets. So, in the additional Y2H-v4 experiment, we only tested pairs that had not been tested in the first experiment. After checking that the scPRS-v2 and scRRS-v2 results were consistent between the two experiments, the overall recovery of AF/RF was calculated combining the data from both experiments, with every pair having been tested in exactly one of the two experiments.

### Y2H-v1 test of AI-predicted human interactome

We tested 4100 randomly sampled pairs from the 2024 bioRxiv version of Zhang et al.[69], along with controls of 250 randomly sampled Lit-BM-24 and 250 random pairs. All were restricted to ORFs in our collection with identical amino acid sequences to the reported UniProt entries. We also included hsPRS-v2 and hsRRS-v2 controls. In analysis these pairs were subsequently filtered for those contained in the 2025 *Science* version of Zhang et al.[3], resulting in 3222 tested pairs.

We pairwise tested these in Y2H-v1[15]. Compared to Y2H-v4, v1 utilizes a different AD vector (pDEST-AD-*CYH2*) and DB vector (pDEST-DB), which are centromeric, low-copy plasmids (Supplementary Table 8). Briefly, each ORF was cherry-picked from our existing human ORFeome collection (hORFeome v9.1), and *MAT*α Y8930:DB-ORF yeast strains (bait) were mated against *MAT***a** Y8800:AD-ORF yeast strains, respectively. Each DB-ORF was also tested against an "AD-null" plasmid without any ORF in the cloning site to control for de-novo autoactivation.

After growing the yeast for 2 days at 30 °C, mating was performed by adding 5 μl AD and 5 μl DB yeast culture into 200 μl YEPD. The next day, 10 μl of mated yeast were transferred into 200 μl SC-Leu-Trp media to select for diploid yeast cells. After growing the cells overnight at 30 °C, 5 μL of the yeast cultures were spotted on selective media (SC-Leu-Trp-His + 1 mM 3AT and SC-Leu-Trp). After growing them for 3 days at 30 °C and 1 day at room temperature, the plates were scored for growth. Spots that did not grow on SC-Leu-Trp were annotated as NA (not successfully tested). Pairs that grew on SC-Leu-Trp-His + 1 mM 3AT were picked into SC-LW to make lysates and perform barcoded 96-well PCR for AD and DB ORFs. To confirm pairs that tested negative (no growth on the 3AT plate), we also sequenced the SC-Leu-Trp plates. Therefore, 10 μL of each well from all fifty 96-well Costar plates were stacked using a 96-well liquid handling robot.

Each well was then purified with the ChargeSwitch™ Plasmid Yeast Mini Kit (ThermoFisher Scientific, CS10203) and PCR amplified with 96-well barcoded AD and DB-forward and barcoded reverse primers. All samples were pooled, purified and sequenced with Nanopore sequencing (Plasmidsaurus). 90% of the positive pairs were sequence confirmed, 83% of the SC-Leu-Trp samples.

### Direct or indirect contact in a complex structure

We queried Interactome3D (version 2020_01)[9] for complexes involving three or more proteins with an experimental structure available.

For all combinations of protein pairs within a complex, the number of residue-residue contacts was calculated with the code and method used in Interactome3D, by accounting for hydrogen bonds, van der Waals interactions, and salt and disulfide bridges. We defined protein pairs with five or more contacts as direct and remaining pairs as indirect. Using this annotation for each dataset, the fraction of direct PPIs was calculated as the number of direct PPIs reported in the dataset divided by the number of direct and indirect pairs reported in the dataset.

### List of genes of unknown function
A list of 979 *S. cerevisiae* genes of unknown function was obtained from Table S9 of Wood et al.[29], of which 950 were in the list of yeast ORFs considered for this study (see section "yeast protein-coding ORFs").

### Publications per gene
Number of publications per gene was extracted from the gene2-pubmed file from NCBI, downloaded on 2018-08-01.

### Domain-domain and domain–motif structural interaction datasets
Domain–domain interactions (DDIs) were taken from the 3DID[75] release 2024-12. Domain–motif interactions (DMIs) were from the ELM[76] interaction set dated 2024-01-30. Both DDI and DMI datasets were then filtered for only PPIs present in I3D-exp. The small number of pairs that were in both the DDI and DMI sets were removed from the DDI set.

### Treatment of heterodimers and homodimers
Unless otherwise noted, homodimers were excluded from most analyses since comparisons between physical interactions and functional relationships are obviously not applicable to single genes (all PCC values of functional profiles would be 1.0 by definition).

### Calculation of recovery rates in Y2H-v4, MAPPIT and GPCA
In MAPPIT and GPCA assays, pairs were scored positive or negative based on thresholds set by the highest-scoring scRRS-v2 pair in the corresponding experiment. For all three assays, pairs without valid quantitative scores were dropped, and recovery rates were calculated as the number of positive pairs over the sum of the positive and negative pairs. The error bars on the recovery rates were calculated using a Bayesian model (a binomial likelihood with a uniform prior), taking the central 68.27% interval of Beta($p+1$, $n+1$), where $p$ and $n$ are the numbers of pairs testing positive and negative, respectively. $P$-values for the difference in recovery between two datasets tested in the same experiment are calculated using Fisher's exact test, two-sided in all cases except when testing a dataset against the scPRS-v2/scRRS-v2 positive or negative controls, where a one-sided test is used. For the yeast AF/RF Y2H-v4 results, where the data were split across two experiments, the scPRS-v2 recovery is calculated as an average of the two experiments, weighting by the number of positive AF/RF pairs in each experiment. For the human AF/RF Y2H-v1 results, sliding window averages were unweighted and used a window size of 250 successfully tested pairs, and error bands were calculated identically to the error bars.

### Calculation of interface areas of PPIs
We retrieved experimental structures using Interactome3D version 2020_05[9]. The interaction interface area was calculated using PyMOL v3.0.0 as half the difference between the solvent accessible surface area of the sum of the isolated individual subunits compared to the complex.

Residue-residue contacts are defined as in Mosca et al.[9]. The definition is:

(i) covalent interactions (disulfide bridges), defined as two sulfur atoms of a pair of cysteines at a distance ≤ 2.56 Å (two times the covalent radius of sulfur plus 0.5 Å);

(ii) hydrogen bonds, defined as all atom pairs N–O and O–N at a distance ≤ 3.5 Å;

(iii) salt bridges, defined as all atom pairs N–O and O–N at a distance ≤ 5.5 Å;

(iv) van der Waals interactions, defined as all pairs of carbon atoms at a distance ≤ 5.0 Å

### Interaction 2D histogram heat maps
For a particular gene/protein property and a network, we ranked all proteins using that property. Tied values were sorted randomly. The proteins were split into an equal number of bins, creating 2D bins of the protein-by-protein space. The number of edges in the diagonal bins was multiplied by a factor of $N^2/(N^2/2 - N/2)$, where $N$ is the number of proteins in the bin, to correct for the smaller number of possible pairwise combinations, since edges were undirected. Homodimeric interactions were excluded.

### Calculation of enrichment for connecting proteins in the same subcellular compartment, pathway, and complex
Subcellular compartment data were obtained from CYCLoPS[77], using the WT data, annotating a protein to a compartment if it has any nonzero value in any of the three repeats. Pathways were obtained from KEGG[78]. Complexes were obtained from CYC2008[79]. The number of PPIs that connected two different proteins in the same compartment, pathway or complex was divided by the mean value for 1000 degree-preserved randomized networks, generated using the Viger and Latapy algorithm implementation through python iGraph v0.11.4[80], and CI values were taken from the innermost 68.27% of the random networks.

### SAFE network visualization
We used the SAFE network visualization tool (v1.5)[28]. The layouts were generated with Cytoscape (v3.4.0)[81] using the edge-weighted spring embedded layout. GO terms were downloaded from the SGD database (version on Jan 17th 2019), and GO[28] terms enriched with $P < 0.05$ were colored and labeled. SAFE analysis was run with the default option except layoutAlgorithm = none (using layout generated by Cytoscape), neighborhoodRadius = 200, and neighborhoodRadiusType = absolute.

### Estimates of the complete yeast interactome size
We used four estimates, relying on partially overlapping assumptions and data, made by independent groups, that predicted the yeast protein binary interactome contains between ~18,000 and ~38,000 direct binary interactions, corresponding to ~0.1–0.2% of all ~18,000,000 possible protein pair combinations[11].

- From Grigoriev et al.[13] 19,500 (15,000–24,000). The reported range was 16,000–26,000, but was calculated using a size of 6300 protein-coding genes, so we adjusted the values using our value of 5883. No central value was reported, and so we used the middle of the range.
- From Yu et al.[11] 18,000 (13,500–22,500 95% CI). Taken from page 107: "we estimated that the yeast binary interactome consists of ~18,000 ± 4500 interactions (SOM VI)" from SOM VI the ± refers to the 95% CI.
- From Stumpf et al.[12] 28,472 (26,650–30,460 95% CI). Taken from the Uetz et al. numbers from Table 1. We use the estimate made using Uetz et al. because three of the other datasets contain indirect protein-protein associations[58,61,64], and the estimate using Ito et al. uses the full dataset, mainly made up of the 'Ito-noncore' subset that was shown to be of poor quality when retested using Y2H and PCA[11].

- From Sambourg and Thierry-Mieg[10] 37,600 (32,252–43,472 95% CI). Taken from page 6: "taken together, this allows us to estimate the size of the binary yeast interactome at ~ 37,600 interactions (95% confidence inter-val: 32252–43472, constructed with the normal approximation method)."

One relatively minor difference between the estimates is that Stumpf et al. are considering only heterodimeric PPIs, whereas Yu et al. and Sambourg et al. are also counting homodimeric PPIs. We account for this when estimating the fraction of predicted inter-actome mapped by excluding homodimers for the Stumpf et al. estimate and including them for the Yu et al. and Sambourg et al. estimates.

The single combined meta-estimate in the main figure was gen-erated using the HKSJ random effects model.

## Prediction of gene functions using guilt-by-association approach

In the guilt-by-association approach, the function of a node (protein) is inferred from the function of its neighbors. In particular, for each node, we count the number of its neighbors annotated with a given function ($n$). This score is then compared to a random benchmark, obtained by randomizing the network 2000 times in a degree-preserved way. Calculating the z-score, $z = (n - \bar{n}) \div \sigma$, is the tradi-tional way of such comparison, obtained by standardizing the original score with the expectation value ($\bar{n}$) and standard deviation ($\sigma$) of the score that would be expected by chance. Yet, the z-score is not free from degree biases and prefers low-degree nodes with extremely small $\sigma$. We therefore apply a related measure, called the effect size. The effect size $n - (\bar{n} + \alpha\sigma)\bar{n}$ is obtained by comparing the original score with the reasonably expected value of the random benchmark, esti-mated as the mean value ($\bar{n}$) and α-times the standard deviation ($\sigma$). In practice, we used $\alpha = 5$, selecting the same candidates as a traditional z-score threshold of $z \geq 5$, but ordering them based on the amount of signal beyond random expectations to avoid a bias towards low-degree nodes. Functional annotations of genes with GO Biological Process terms were obtained as described above and further restricted to annotations with the experimental evidence codes EXP, IDA, IPI, IMP, IGI, IEP, HTP, HDA, HMP, HGI, and HEP.

## Network fragmentation

The error bands are generated from randomly sampling a fraction of the PPIs from each network, where the fraction varies from 5 to 95% in 5% increments, with 1000 random subnetworks generated at each point.

## Degree distribution plots

Degree distributions were plotted according to Chapter 4, Advanced Topic 3.B of Network Science[82], on a log-log scale with logarithmic binning, with the unbinned data shown in grey.

## Thresholds for AF/RF confidence based on Y2H test results

For the yeast AF/RF, which was generated using a single threshold, the predicted pairs were divided into six bins of contact probability of roughly equal size, and the -core cutoff was defined as the start of the first bin whose Y2H-v4 positive rate was not significantly lower than that of scPRS-v2, which was 0.95. For the human AF/RF, which was generated using multiple thresholds, the positive rate was calculated across a sliding window of 250 pairs, separately for each strategy, using the corresponding score for each strategy, and the threshold was defined as the low end of the lowest score window to not be below the lower bound of the confidence interval on the Lit-BM-24 positive rate. The derived thresholds were 0.9696 for strategy 3 (genetic interac-tions → RF2-PPI), 0.9787 for strategy 5 (de novo screen on pairs with

common cellular compartments → AF2), and 0.9800 for strategy 7 (genetic interactions → RF2-PPI → AF2).

## Observation in previous datasets

Experimental structures were defined as I3D-exp-24 (yeast) I3D-exp-20 (human). Interolog (homologous PPI) structures were defined by first performing a sequence alignment of all translated ORFs from SGD (yeast) or the UniProt reference proteome UP000005640_9606 (human) against the a.a. sequences in the PDB (https://ftp.ncbi.nlm.nih.gov/blast/db/FASTA/pdbaa.gz downloaded 2024-02-06), using mmseqs with a E-value threshold of $10^{-5}$, then any PDB structure where any two different chains were matched to the two different interacting proteins was considered an interolog. High-quality binary PPIs referred to ValBin-24 (yeast) and Lit-BM-24 + HI-union (human). Co-complex associations referred to all other literature data for biophysical inter-actions and presence within the same complex from the EBI complex portal (release 2026-01-09). For AF/RF-core (yeast), the previous datasets were restricted to those published before its publication date of 2021-11-11.

## AlphaFold structure predictions

Predicted structures for pairs in YeRI, scPRS-v2, and scRRS-v2 were generated with AlphaFold version 2.3.1[5]. With options:

```
--model_preset=multimer
--db_preset=full_dbs
--max_template_date=2023-05-05
--num_multimer_predictions_per_model=1
--enable_cpu_relax
--use_precomputed_msas.
```

We could only predict structures for the pairs whose combined number of amino acids were 2500 or less.

We extracted the PAE matrix from the AlphaFold pickle output using the alphapickle script v1.4.1 (https://github.com/mattarnoldbio/alphapickle DOI:10.5281/zenodo.5752375). We calculated five different confidence metrics for each structure

1. Model confidence (0.8 ipTM + 0.2 pTM)
2. Contact probability (max probability of a pair of residues between the two proteins <12 Å)[2]
3. pDockQ (logistic function of interface plDDT and number of contacts)[40]
4. iPAE[41] (median interface PAE)

## CCC (number of residue-residue contacts with PAE < 4 Å)

Each putative residue-residue contact in an AlphaFold-multimer pre-dicted structure also has a corresponding PAE in Angstrom.

AlphaFold3 predictions were obtained from the AlphaFold Server[6].

## Confident contacts count (CCC)

We designed the CCC metric with the expectation that it should be more robust than the contact probability because it is not driven by only a single pair of high-confidence proximal residues. Addition-ally, CCC should avoid the issue with most alternative metrics, which, by averaging the confidence scores of all interface residues, can miss a high-confidence interface if it is obscured by another interface predicted with low confidence. The 4 Å cutoff on PAE was an arbitrary choice, reasoned to be around the length of the non-covalent bonds.

## Precision–recall curves

We calculated balanced precision and plotted monotonized precision–recall curves as described by Wu et al.[83].

## PPIs with no structural model

Yeast PPIs with no structural model were defined as not being present in I3D-exp-24, not having an interolog structure (defined above), and not being in the full AF/RF dataset (i.e., the dataset before applying the higher, experimentally-derived contact probability threshold).

## Reporting summary

Further information on research design is available in the Nature Portfolio Reporting Summary linked to this article.

## Data availability

Protein interaction data have been submitted to the IMEx consortium (http://www.imexconsortium.org) through IntAct[57] and assigned the identifier IM-30553. Predicted structures of YeRI PPIs are deposited at https://doi.org/10.5281/zenodo.18601049. YeRI, Y2H-union-25, and ValBin-25 maps are available at our OpenPIP[84] website: https://yeast.interactome-atlas.org/. Source data are provided with this paper.

## Code availability

Analysis code is available at https://github.com/ccsb-dfci/ai-interactome-experimental-assessment, archived together with the input data at https://doi.org/10.5281/zenodo.18499797.

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

## Acknowledgements

We thank Steffi de Rouck for help with the MAPPIT experiments. We thank Gary Bader and acknowledge past and current members of the Center for Cancer Systems Biology (CCSB) for helpful discussions and experimental help. We thank Qian Cong and Jing Zhang for providing access to additional data files related to their AI PPI predictions. This work was funded by the following sources. National Institutes of Health grant R01HG006061 (M.V., D.E.H., M.A.C., M.E.C., and P.F.-B.). National Institutes of Health grant R01GM130885 (M.V.). National Institutes of Health grant R01GM133185 (M.V., M.A.C., and F.P.R.). Institute Sponsored Research funds from the Dana-Farber Cancer Institute Strategic Initiative (M.V.). Canadian Institutes of Health Research (CIHR) Foundation Grant FDN159926 (F.P.R.). Canadian Institutes of Health Research (CIHR) Project Grant PJT-162410 (J.R.). Léon Fredericq Foundation (A.D. and F.L.). Fund for Scientific Research (FRS-FNRS) Télévie Fellowships #7651317 F (A.D., J.-C.T.) and #7459421F (F.L., J.-C.T.). Natural Sciences and Engineering Research Council (NSERC) of Canada Banting Postdoctoral Fellowship (D.-K.K.). National Research Foundation (NRF) of Korea Basic Science Research Program grant 2017R1A6A3A03004385 funded by the Ministry of Education (D.-K.K.). National Institutes of

Health National Resources For Network Biology (NRNB) Google Summer of Code 2015 (M.W.M.). U.S. National Science Foundation PHY-2440223 POLS NSF CAREER Award sponsored by NSF 22-586, and by the NSF–Simons National Institute for Theory and Mathematics in Biology, jointly funded by the U.S. National Science Foundation DMS-2235451 and the Simons Foundation MP-TMPS-00005320 (I.A.K.). Spanish Ministry of Science Ramon y Cajal fellowship RYC-2017-22959 (C.P.). Dana-Farber Cancer Institute Center for Cancer Systems Biology (CCSB) Deborah F. Allinger Fellowships (A.Y., L.L.). Belgian American Educational Foundation (BAEF) Doctoral Research Fellowships (F.L.). Wallonia-Brussels International (WBI)-World Excellence Fellowships (F.L.). Herman-van Beneden Prize (F.L.). Josée and Jean Schmets Prize (F.L.). M.V. is a Chercheur Qualifié Honoraire, and J.-C.T. is a Directeur de Recherche from Fonds de la Recherche Scientifique (FRS-FNRS, Wallonia-Brussels Federation, Belgium). Free State of Bavaria's AI for Therapy (AI4T) Initiative through the Institute of AI for Drug Discovery (AID) (P.F.-B.) and the Impuls and Networking Fund of the Helmholtz Association (PhenoPred) (P.F.-B.). J.D.L.R. acknowledges a Fulbright Grant Senior Scholar Grant (ref. PRX23/00628) awarded to work in the CCSB of the DFCI from January to July 2025.

## Author contributions

Computational analyses were performed by L.L., Y.W., with help from B.C., D.D.R., T.R., K.L., and O.D. Interactome mapping experiments were performed by A.D., T.C., with help from S.S., N.J., Q.Z., Z.Y., and K.S.-F. Sequencing to identify interacting proteins was carried out by A.G.C., M.G., N.K., J.J.K., and J.C.M. Y2H vectors were designed and generated by Q.Z. with help from N.J. The preparation of Y2H, GPCA, and MAPPIT destination clones by en masse gateway cloning and yeast transformations were performed by Q.Z., N.J., A.D., and T.C. Experimental results were processed by Y.W., T.H., and K.L. GPCA validation experiments were done by A.D., T.C., with help from Y.J. MAPPIT validation experiments were done by I.L., supervised by J.T. Y2H tests of predicted yeast interactions were performed by K.S.-F. Y2H tests of predicted human interactions were performed by F.L. and K.S.-F. with help from G.G.M. Functional enrichment analyses were done by D.-K.K., L.L., and Y.W. Extraction of the literature datasets was performed by L.L., T.H. YeRI web portal was built by M.W.M., supervised by J.R., M.H. Structural analyses were done by C.P., L.L., and Y.W., supervised by P.A. Images of 3D structural PPI models were produced by J.D.L.R. Topological analyses were done by L.L. Sequencing analyses were done by T.H., W.B., Y.S., and Y.W. Network-based functional prediction was performed by I.A.K. Additional experiments were performed by F.L., V.V.B.J., and G.M. The overall research effort was designed and conceptualized by M.V., F.P.R., M.A.C., D.E.H., P.F.-B., Y.W., A.D., L.L., and A.Y. Interactome mapping was supervised by B.C., M.V., M.A.C., D.E.H., and T.H. Manuscript was written and edited by L.L., A.Y., Y.W., A.D., T.H., F.L., F.P.R., J.D.L.R., P.F.-B., D.E.H., M.A.C., J.-C.T., and M.V. with contributions from other co-authors. The overall research effort was supervised and/or advised by M.V., F.P.R., M.A.C., and D.E.H. The project was conceived by M.V. Major funding acquisition was by M.V., D.E.H., M.A.C., F.P.R., P.F.-B., and M.E.C. D.-K.K., F.L., K.S.-F. contributed equally and should be considered co-second authors.

## Competing interests

J.C.M. is a founder and CEO of seqWell, Inc; F.P.R., M.V. are shareholders and scientific advisors of seqWell, Inc. J.-C.T. is a founder of ExtraCell Biotech, SRL. The remaining authors declare no competing interests.

## Additional information

**Supplementary information** The online version contains Supplementary material available at https://doi.org/10.1038/s41467-026-70942-x.

Luke Lambourne [ID] [1,2,3,30], Anupama Yadav[1,2,3,30], Yang Wang[1,2,3,30], Alice Desbuleux[1,2,3,4,30], Dae-Kyum Kim[1,5,6,7], Florent Laval [ID] [1,2,3,4,8], Kerstin Spirohn-Fitzgerald [ID] [1,2,3], Tiziana Cafarelli[1,2,3], Carles Pons[9], István A. Kovács[1,10,11,12,13], Noor Jailkhani[1,2,3], Sadie Schlabach[1,2,3], David De Ridder[1,2,3], Katja Luck [ID] [1,2,3], Vladimir V. Botchkarev Jr.[1,2,3], Olivia Debnath[1,2,3], Wenting Bian[1,2,3], Yun Shen[1,2,3], Zhipeng Yang[1,2,3], Miles W. Mee[5,14], Mohamed Helmy[5], Yves Jacob[1,15,16], Irma Lemmens[17,18], Thomas Rolland [ID] [1,2,3], Gregory G. McClain[1,2,3], Atina G. Coté [ID] [1,5,6,7], Marinella Gebbia[1,5,6,7], Nishka Kishore[1,5,6,7], Jennifer J. Knapp[1,5,6,7], Joseph C. Mellor[1,5,6,7,19], Gonen Memisoglu [ID] [20], Jüri Reimand [ID] [6,14,21], Jan Tavernier[17,18], Michael E. Cusick[1,2,3], Quan Zhong [ID] [1,2,3,22], Patrick Aloy [ID] [9,23], Tong Hao[1,2,3], Benoit Charloteaux [ID] [1,2,3], Frederick P. Roth[1,5,6,7,24], Javier De Las Rivas [ID] [25,26], Pascal Falter-Braun [ID] [1,2,3,27,28] ✉, David E. Hill [ID] [1,2,3] ✉, Michael A. Calderwood [ID] [1,2,3] ✉, Jean-Claude Twizere [ID] [1,4,8,29] ✉ & Marc Vidal[1,2,3] ✉

¹Center for Cancer Systems Biology (CCSB), Dana-Farber Cancer Institute, Boston, MA, USA. ²Department of Genetics, Blavatnik Institute, Harvard Medical School, Boston, MA, USA. ³Department of Cancer Biology, Dana-Farber Cancer Institute, Boston, MA, USA. ⁴Laboratory of Viral

Interactomes, Computational and Molecular Biology Unit, GIGA Institute, University of Liège, Liège, Belgium. [5]Donnelly Centre for Cellular and Biomolecular Research (CCBR), University of Toronto, Toronto, ON, Canada. [6]Department of Molecular Genetics, University of Toronto, Toronto, ON, Canada. [7]Lunenfeld-Tanenbaum Research Institute (LTRI), Sinai Health, Toronto, ON, Canada. [8]TERRA Teaching and Research Centre, University of Liège, Gembloux, Belgium. [9]Institute for Research in Biomedicine (IRB Barcelona), The Barcelona Institute for Science and Technology, Barcelona, Spain. [10]Network Science Institute, Northeastern University, Boston, MA, USA. [11]Department of Physics and Astronomy, Northwestern University, Evanston, IL, USA. [12]Northwestern Institute on Complex Systems, Northwestern University, Evanston, IL, USA. [13]NSF-Simons National Institute for Theory and Mathematics in Biology, Chicago, IL, USA. [14]Computational Biology Program, Ontario Institute for Cancer Research, Toronto, ON, Canada. [15]Unité de Génétique Moléculaire des Virus à ARN (GMVR), Département de Virologie, Institut Pasteur, UMR3569, Centre National de la Recherche Scientifique (CNRS), Paris, France. [16]Université Paris Diderot, Paris, France. [17]Center for Medical Biotechnology, Vlaams Instituut voor Biotechnologie (VIB), Ghent, Belgium. [18]Cytokine Receptor Laboratory (CRL), Department of Biomolecular Medicine, Faculty of Medicine and Health Sciences, Ghent University, Ghent, Belgium. [19]seqWell, Beverly, MA, USA. [20]Department of Molecular Genetics and Cell Biology, The University of Chicago, Chicago, IL, USA. [21]Department of Medical Biophysics, University of Toronto, Toronto, ON, Canada. [22]Department of Biological Sciences, Wright State University, Dayton, OH, USA. [23]Institució Catalana de Recerca i Estudis Avançats (ICREA), Barcelona, Spain. [24]Department of Computational and Systems Biology, University of Pittsburgh, Pittsburgh, PA, USA. [25]Cancer Research Center (CiC-IBMCC, CSIC/USAL), Consejo Superior de Investigaciones Científicas (CSIC)/University of Salamanca (USAL), and Instituto de Investigacion Biomedica de Salamanca (IBSAL), Salamanca, Spain. [26]Institute for Biomedical Research of Salamanca (IBSAL), Salamanca, Spain. [27]Institute of Network Biology (INET), Molecular Targets and Therapeutics Center (MTTC), Helmholtz Munich, German Research Center for Environmental Health, Neuherberg, Germany. [28]Microbe-Host Interactions, Faculty of Biology, Ludwig-Maximilians-Universität München, Planegg-Martinsried, Germany. [29]Division of Science and Math, New York University Abu Dhabi, Abu Dhabi, UAE. [30]These authors contributed equally: Luke Lambourne, Anupama Yadav, Yang Wang, Alice Desbuleux. ✉e-mail: pascal.falter-braun@helmholtz-munich.de; david_hill@dfci.harvard.edu; michael_calderwood@dfci.harvard.edu; jean-claude.twizere@uliege.be; marc_vidal@dfci.harvard.edu

