## [Transparent Peer Review File · Nature Communications]

Experimental assessment of AI-based interactome mapping

Corresponding Author: Professor Marc Vidal

Version 0:

Reviewer comments:

Reviewer #1

(Remarks to the Author)

Report on Lambourne et al. « Experimental assessment of AI-based interactome mapping »

Lambourne and team provide a straight forward (in the sense of well written and illustrated) comparison of two AF complex prediction data set, one from yeast and one from human (AF/RF, still preprint), with experimental Y2H PPI data. Intriguingly, the (novel and interesting) outcomes from the comparisons for yeast and human are very, very similar for both species: The predictions are of high quality, they can be experimentally corroborated to a similar rate than known positive reference PPIs. There are no hubs, the AF predicted networks appear disintegrated, likely because PPIs with IDPs are missing. Interfaces are larger, domain-domain PPIs are predominant, there is a higher knowledge bias, and most importantly the AF predictions contribute very few novel PPIs only. Experimental PPI discovery is warranted. Along this conclusion, a new Y2H yeast PPI data set is presented, high quality. It is derived from the all vs all Y2H mapping of 5854 ORFs and adds 1880 PPIs. The combined ValBin-25 PPI yeast dataset has resource character. The study ends by using AF to predict the 3D structure of reference pairs and novel PPIs to demonstrate how AF/RF can contribute to interactome research. In summary, this work is a significant contribution!

Points for consideration.

- a) The authors use a new Y2H-v4 vector setup which it is claimed to be an orthogonal system with respect to previous screens. However, the screen delivers very sparse data. Considering the high sensitivity (22%, 1d), I do not see how the data go along with interactome size predictions. In Figure 1J the val-bin24/25 is between 25-50% of the interactome size and counts about 10000. The screen covers roughly all pairs (“a foundation for the systematic completion of the *S. cerevisiae* interactome map”) and detects 1880/0.22= 8545 PPIs. Please, consider that the sensitivity using the PRS is a substantial overestimate.
- b) As mentioned above, the study ends by using AF to predict the 3D structure of reference pairs, 37% success and some novel PPIs with 14% success. The authors provide hypothesis for the difference, which is not reflected in the experimental validation. While I go well along with the offered explanations, the authors could add at least two more limitations to the analysis. This is the PRS is more like the domain-domain AF data, or at least theoretically the new yeast data are not as reliable. What is the average surface area of RPS pairs etc ...
- c) After detailed analysis of yeast data (figs 1-3) the authors perform the analyses of human data, with analogous outcome. While these results are very well supported and thus convincing, the human time line is not. In analogy, the human time line (Figure 4d) should include (similar to yeast Figure 3i), also the non-Vidal-Lab large scale data sets that contributed to the binary interactome mapping. The Vidal lab contributed the most significant data sets, but not exclusively. Including the most significant data sets also from other labs will strengthen the point of the Figure.
- d) It should be note that while I strongly agree that the novel PPIs in the human AF paper (which I have read carefully) are very few in number, they mostly relate to membrane-membrane PPIs, where Y2H is negatively biased. Therefore, the novel AF PPIs could be largely complementary to the experimental binary PPIs. This needs to be checked e.g. in terms of which proteins are involved etc. How many of the strictly novel PPIs of the human AF/RF core have been retested with Y2H-v4 in the presented experiments? I suggest a similar panel like the ones in Figure 4a that show the success of de novo screens would be revealing if shown for strictly novel PPIs (provided that they were tested).
- e) The predictions of novel Y2H were done with success even though the pairs were not in the AF predicted data set (maybe different scoring, ccc). However, where these novel pairs considered in the AF predictions in the first place? Did they end up below cut off (cf . Figure 4c). The human predictions were guided by known pairs though ...

Reviewer #2

(Remarks to the Author)

The manuscript by Lambourne et al consists of three parts. In the first, the authors assess the current state of interactome mapping for budding yeast. Using three binary interaction assays, they test subsets of pairs from various interaction datasets to evaluate their quality. Most of the interactions are retested using an updated version of the Vidal lab yeast two-hybrid (Y2H) pipeline that is able to positively identify more interactions (higher sensitivity) while maintaining a low false-positive rate. The main conclusion is that most datasets are of low quality in terms of accurately reporting direct protein-protein interactions, possibly in part because many datasets report indirect associations (i.e. proteins in a larger complex) in the form of interacting pairs. The authors combine the high-quality datasets into a single dataset they term ValBin-24 for validated binary interactions in 2024. This section provides a thorough and thoughtful overview of the current state of interactome mapping.

In the second part of the manuscript, the authors experimentally expand the yeast binary interaction map using i) their updated Y2H pipeline, and ii) a near complete yeast ORFeome collection generated here by the authors (adding 921 ORFs to an existing collection of 4933 ORFs). This adds 1723 interactions to previous Y2H high confidence interactions, for a total of 4556 interactions. Both the expanded ORFeome collection and new interactions are of high value to the scientific community.

In the third part, the authors assess the accuracy of yeast and human protein interactions predicted by new structure prediction algorithms. They do this by testing all predicted yeast interactions and a random sampling of human protein interactions in their Y2H pipeline. I found this an interesting analysis to read. The main conclusion is that while predictions are of high quality, very few new interactions are predicted. The likely reasons for this are that structural prediction algorithms are better at prediction structures for proteins similar to proteins with known structures, and are not good at identifying interactions with disordered regions, which mediate many protein interactions. While these observations are not necessarily unexpected, it is very useful to have a thorough examination of the predictive power of new AI methods.

Overall, the work provides an advancement in our knowledge of protein-interactions and in tools (Y2H system and new ORFeome dataset). The evaluation of AI methods should be of broad interest and is an important step in assessing the use cases for these new tools.

I have little if any qualms with the experimental approaches used. The Vidal lab are absolute experts in the field of binary protein interaction mapping, having honed their Y2H pipeline over 2 decades of work and having increasingly emphasized the value of rigorous benchmarking of interaction assays with positive and negative reference sets. The comparisons with AlphaFold predictions also seem to have been thoroughly performed, with attention to the computational methods and thresholds used. My comments for improvement are below, most address readability.

- A table with all the various datasets mentioned would be very helpful, even being familiar with past work from the Vidal lab the many dataset abbreviations confused me. Most are explained in the methods, but a overview table to refer to while reading would be a welcome addition.

- The multiple Lit-BM sets are not well explained. On page 4 line 28, the authors state that interactions from Lit-BM-17 were tested, but on line 32 it seems a Lit-BM-13 was also tested. What is the difference?

- On page 5 line 26, why are the new interactions compared to Lit-BM-13? In figure 1, Lit-BM-17 was used.

- Page 5 line 36: the sentence 'thus we generated a new high quality map' would be better placed on line 23, before going on to evaluating the new map. The way it reads now I was wondering if the new map was generated using criteria based on the comparison with the PSNs, e.g. only incorporating those that were enriched in one or more PSNs. (or if that is actually the case, clarify the criteria).

- The naming of the Yeast Reference Interactome seems an odd choice. What makes this a reference interactome, why would CCSB-YI1 not be a reference interactome, or the next interactome the Vidal lab generates. Wouldn't a year name be more consistent with their nomenclature, e.g. CCSB-YI25? Or YI2? The analyses below are also not done on YeRI but on union-25 and on ValBin-25, so it isn't used as a reference set here either.

- The combining of YeRI with 'the datasets previously identified as high-quality binary interactions' wasn't clear to me. Do the authors mean ValBin-24? I also don't think that a table of ValBin-25 with the source(s) of each interaction is included.

- On page 5, the authors note that the Y2H system stands out for having contributed large numbers of reliable and primarily direct PPIs. If the authors want to make a point that Y2H is a better system than others, this statement needs more substantiation. Specifically, I think there may be a circular argument here for 2 reasons: 1) most available binary interactions were identified by Y2H, and 2) in the retesting, most interactions were retested by Y2H thus having a higher chance to retest positively than interactions identified by other methods. While most high quality direct interactions may stem from the Y2H system, this may therefore not be of particular note and not reflect an intrinsic higher quality of the Y2H system.

- The guilt-by-association experiments are difficult to interpret in their current form. The example in note 2 is clear, but the experiment is not. I understand that the authors first assign a potential function based on potential functions of its neighbour (GO term based). Then, during the time between the analysis and paper writing, some proteins went from unknown to studied, allowing a determination of whether that assessment was correct. So I was expecting table 18 to have a list of proteins, a predicted function, and an assessment whether new information shows if that prediction is right. But that is now what table 18 shows. How should the reader interpret that table? Their example YGR168C is listed 86 times, so apparently 86 possible functions were assigned? And how do I see from the table that that assignment was correct?

- Page 6 line 19 'testing all 1505 pairs'. This sounds like a new experiment, but this is just data taken from YeRI correct? (Since that experiment tested all pairwise combinations).

- Page 7 line 11: how do the strictly novel predicted PPIs fare in the Y2H retest?
- Page 7 line 39: why was Y2H-v1 used to retest the predicted human interactions?
- Page 9 line 38: shouldn't ValBin-25 be the new gold-standard?
- Page 10 line 14/15. If AlphaFold can predict models for interactions not detected in the interaction screen, does that mean that the computational prediction pipeline can be improved and identify more of the missing interactions?
- Page 10 line 20: typo in 'could be may have been'
- Supplementary info: in generating the new negative reference set the authors filter out any pair that appeared in any experimental dataset. While a logical approach, I wonder if at some point you start to bias the negative reference set for protein pairs that are particularly unlikely to be detected as false positives, for example because they have overall opposing charges or structures that are not likely to be 'sticky'. In theory you want to test true random protein pairs but this approach is no longer truly random. Perhaps only validated direct protein interactions should be subtracted.

Reviewer #3

(Remarks to the Author)

On the manuscript entitled "Experimental assessment of AI-based interactome mapping", Vidal and co-authors present a comprehensive effort to evaluate the performance of AlphaFold in global deciphering of quaternary structures (protein-protein interactions, PPIs) in yeast and human, and at the same time provide the community with a new experimental method for detection of such interactions that provide a large number of novel high-quality PPIs. The work is thorough and sound. The presented study is endowed with multiple virtues and contributions, among which the critical assessment of the usability of PPI data is not the least. Overall, the raised claims, the presented results and the contextual discussion correspond well with each other. The manuscript is well-written (although sometimes a bit difficult to read, due to the lack of space and the need to consult with the profuse additional materials). In my opinion, the manuscript would be of the highest interest in PPI research and may also appeal to the wider community, and has my enthusiastic support for its publication in Nature Communications.

There are a few aspects that, in my opinion, could make the manuscript stronger, however. I will list them below for the Author's consideration:

1. One of the main contributions of the manuscript (highlighted at the very end of the discussion) is the new YeRI dataset, providing over 1400 novel protein interactions in yeast. This is achieved by implementing a variation of the Yeast-two-Hybrid (Y2H) complementation assay, aptly named Y2H-v4 in the manuscript. For the benefit of the wider audience, I think the manuscript would benefit from a more thorough explanation of the technique and an explanation of its similarities and differences with the previous versions of Y2H. Ideally, Fig.1 (and failing to that, the corresponding supplementary information) could allocate these explanations.
2. The Authors mention a large number of existing datasets, especially in the first part of the manuscript (yeast). A graphic navigating the reader through those datasets would be extremely useful, either in the main manuscript or in its accompanying supplementary information. Such a guide could include the nature of each of the sets, their size, and the reason for being used in the current study.
3. Following the previous point, the Authors define Y2H-union-25 as the merging of their YeRI with three previously existing datasets, comprising a total of 4,307 PPIs. I had issues identifying which previously existing datasets the Authors refer to. I guess these are refs. 11, 18, and 19, but clarification on the manuscript would be nice. On the same tone, on pg. 6, the Authors state that adding YeRI information to "previously identified as high-quality binary interactions results in a ValBin-25 dataset of 10,759 PPIs." It would be valuable if the Authors could identify (refer to) those previously described high-quality binary interactions.
4. A key question that is only partially addressed during the text is the true complementarity or orthogonality nature of different methods. I will illustrate this with the following case. YeRI detects around 1900 PPIs (of which more than 1400 novel) while the estimates for the total yeast interactome range between 20 and 40 thousand (pg. 3). The Authors, later by integrating YeRI to Y2H-union24 and further to ValBin25, increase the coverage to around 10 thousand. The open questions here are twofold: what are the remaining (10-30K) yeast PPIs? How can a naive user remove older datasets containing incorrectly recorded PPIs? The Authors address the latter question in the second part of the manuscript (human), stating that various thresholds on different estimation techniques can be applied (Fig. 4a), but a clearer guideline on what can be considered a wrong PPI would be useful to the readers.
5. On page number 8, the Authors state that the human AF/RF-core dataset consists of 87% of the predicted PPIs. This number is (strikingly?) different to that obtained for yeast (64%, pg. 6). Could the Authors comment if these differences are significant and, if so, what are their plausible origins?
6. On the side of minor notes:
 - 6.1. At the beginning of the manuscript, the Authors refer systematically to the positive reference set as scPRS-v2. As the manuscript develops, this is sometimes substituted by PRS. Unity on naming would help the reader to understand that the Authors are referring to the same dataset.
 - 6.2. Despite the amazing effort in keeping the text short and clear, I think the Authors have some editing errors on page 10,

line 20. The text reads “[...] some of the interactions for which structures could be may have been missed [...]”. I think the current grammatical formulation is incorrect. Instead, “could have been missed” or “may have been missed” sounds better.

Version 1:

Reviewer comments:

Reviewer #1

(Remarks to the Author)

Lambourne et al. addressed the points raised and amended text and adapted figures and refs. They also updated their human interactome analysis in the light of the recently published interactome prediction paper by Zhang and colleagues. I want to congratulate the authors to their work and suggest to go ahead.

(Remarks on code availability)

Reviewer #2

(Remarks to the Author)

The authors have extensively revised the manuscript and I appreciate the well written rebuttal, as well as their update of the analysis of the Alpredicted human protein-protein interactions to the published version of that dataset. All of my points have been thoroughly addressed.

(Remarks on code availability)

Reviewer #3

(Remarks to the Author)

In my opinion, the Authors satisfactorily answered all reviewers' questions. I think the manuscript is ready for publication.

(Remarks on code availability)

**Response to the reviewers for the manuscript:
“*Experimental assessment of AI-based interactome mapping*”**

We thank the reviewers for their thorough and constructive evaluation, which included both suggestions for additional analyses and conceptual questions as well as requests for clarification. In response, we have performed such analyses and addressed all their questions and requests.

In addition, we took the initiative to address an issue that arose since we received our reviews. Indeed, the bioRxiv preprint (bioRxiv 2024.10.01.615885) describing AI-predicted human protein-protein interactions (PPIs), originally experimentally assessed in our manuscript, has now been published (Zhang *et al. Science* 2025). And we noticed that the final dataset published in *Science* is different from the one described in the bioRxiv version. By rerunning our analyses on the *Science* version of the predicted dataset, we verified that this update did not affect our conclusions.

The manuscript has been revised to incorporate these new analyses and corresponding discussion. Importantly, all additional analyses support our original conclusions.

We hope that the editor and reviewers will agree that these revisions have clarified and substantially strengthened the manuscript. Below, we address each comment in turn and describe how the revised manuscript has been modified since the previous submission.

AUTHOR NOTE TO THE REVIEWERS AND THE EDITOR

The set of human predicted PPIs we tested in our original submission was taken from a bioRxiv preprint (bioRxiv 2024.10.01.615885). After we received the reviewers' comments, a revised version of this preprint was published in *Science* on September 25, 2025, accompanied by an updated version of the dataset. To determine whether the updates introduced in the published dataset affected the conclusions drawn from our experiments and analyses based on the preprint data (see Figure below), we systematically examined the differences between the preprint and published versions.

The overlap of predicted human PPIs between the preprint and published datasets.

Importantly, all algorithms and input data (*i.e.*, the version of the PDB, the multiple sequence alignments, and the versions of the PPI and genetic interaction databases) remained unchanged between the preprint and published versions. The differences arise from two main changes to the methodology. First, the published version introduces an additional filter for homologous protein pairs, which the authors determined reduced false positive predictions. This change accounts for predicted PPIs present in the preprint dataset but absent from the published dataset. Second, input pairs with prior support from PPI databases were subdivided according to their source database and the amount of supporting evidence, with separate thresholds derived for each subset. This change accounts for predicted PPIs included in the published version that were not present in the preprint.

In our updated analysis, we were able to derive thresholds for the *de novo* screening and genetic interaction strategies using our experimental data by filtering out these homologous pairs. However, we were not able to accurately derive thresholds for the new PPI DBs strategies (*i.e.*, the subset of predictions where pre-existing PPI evidence is used) based on the random sample of the preprint dataset that we tested. However, given that these pairs have been observed previously, we think their evaluation is significantly less critical to our findings than the experimental evaluation of pairs that emerged from the *de novo* screen.

We have updated our manuscript by restricting the analyses to the 17,849 pairs in the dataset published in *Science*. After filtering out the predicted PPIs from the original bioRxiv version that do not now pass the adjusted thresholds of the published version, the number of pairs we tested in Y2H decreased from 4,046 to 3,222. Importantly, the values of the thresholds that we derived using our approach were highly similar between

the preprint and published versions of the dataset, and our conclusions remain unchanged. This can be seen in the before and after versions of Figure 4 below.

Fig. 4

Previously submitted version of Figure 4, based on the bioRxiv dataset.

Fig. 4

Updated version of Figure 4, filtered for predicted PPIs in the published *Science* dataset.

Reviewer #1 (Remarks to the Author)

Report on Lambourne et al. Experimental assessment of AI-based interactome mapping

Lambourne and team provide a straight forward (in the sense of well written and illustrated) comparison of two AF complex prediction data set, one from yeast and one from human (AF/RF, still preprint), with experimental Y2H PPI data. Intriguingly, the (novel and interesting) outcomes from the comparisons for yeast and human are very, very similar for both species: The predictions are of high quality, they can be experimentally corroborated to a similar rate than known positive reference PPIs. There are no hubs, the AF predicted networks appear disintegrated, likely because PPIs with IDPs are missing. Interfaces are larger, domain-domain PPIs are predominant, there is a higher knowledge bias, and most importantly the AF predictions contribute very few novel PPIs only. Experimental PPI discovery is warranted. Along this conclusion, a new Y2H yeast PPI data set is presented, high quality. It is derived from the all vs all Y2H mapping of 5854 ORFs and adds 1880 PPIs. The combined ValBin-25 PPI yeast dataset has resource character. The study ends by using AF to predict the 3D structure of reference pairs and novel PPIs to demonstrate how AF/RF can contribute to interactome research. In summary, this work is a significant contribution!

AUTHOR RESPONSE

We thank the reviewer for this positive summary of our work.

Points for consideration.

*a) The authors use a new Y2H-v4 vector setup which it is claimed to be an orthogonal system with respect to previous screens. However, the screen delivers very sparse data. Considering the high sensitivity (22%, 1d), I do not see how the data go along with interactome size predictions. In Figure 1J the val-bin24/25 is between 25-50% of the interactome size and counts about 10000. The screen covers roughly all pairs (“a foundation for the systematic completion of the *S. cerevisiae* interactome map”) and detects $1880/0.22= 8545$ PPIs. Please, consider that the sensitivity using the PRS is a substantial overestimate.*

AUTHOR RESPONSE

We agree with the reviewer that the assay sensitivity estimated using the PRS does not entirely reflect the overall sensitivity of the YeRI dataset. As demonstrated by our group (Yu et al. *Science* 2008 and Venkatesan et al. *Nature Methods* 2009), another important known factor affecting the ability of YeRI to approach the theoretical assay sensitivity, as measured by the PRS, is the sampling sensitivity, which quantifies the saturation of

screens relative to the pairwise tests used in the benchmarking. The sampling sensitivity from Yu *et al. Science* 2008 and Venkatesan *et al. Nature Methods* 2009 was around 80% after three screens. However, those experiments were performed using pool sizes of 188 ADs. In contrast, YeRI was mostly generated with pool size of 700 ADs, and we know, based on unpublished internal data, that an increasing pool size negatively affects the sampling sensitivity, i.e., decreases the saturation of a single screen. Therefore, we expect a correspondingly lower sampling sensitivity, which would explain a large part of the discrepancy between your value of 8,545 PPIs and the interactome size estimates.

If we conservatively estimate the sampling sensitivity to be 50%, instead of the 80% previously determined for the smaller pools, applying the empirical framework to Y2H-v4 would yield an estimate of ~17,000 interactions. This is in line with two of the four previous estimates we use in the manuscript, one of which used the same methodology on a different dataset (Yu *et al. Science* 2008), the other used a completely different methodology (Grigoriev *NAR* 2003).

At the same time, the assay sensitivity estimated from ~100 pairs is associated with an error margin. In addition, we agree that it is possible that the PRS is biased, as we rely on the existing literature for assembling this high-quality benchmark set. As a reminder, to serve as a widely acceptable positive control set, the PRS is composed of interaction pairs that are well-established and validated by multiple studies, both biophysically and functionally. Thus, biases will be introduced by the focus of the existing literature and by the need to only include interactions that have been reproduced by the community. It is important to point out that no additional filters are applied, meaning we do not remove interaction pairs that are dependent on post-translational modifications or involve transmembrane proteins. We consider the difficulties that any assay (including the Y2H) may have with such 'special' pairs as part of the assay's inherent limitations that we aim to capture with the assay sensitivity as determined by the PRS. We've added this clarifying statement to the main text at the end of the second section describing the generation of YeRI:

EXCERPT FROM REVISED MANUSCRIPT

"Notably, the overall sensitivity of YeRI is lower than the 22% scPRS-v2 recovery (Fig. 1d) due to the 'sampling sensitivity'³⁰ of the screen."

b) As mentioned above, the study ends by using AF to predict the 3D structure of reference pairs, 37% success and some novel PPIs with 14% success. The authors provide hypothesis for the difference, which is not reflected in the experimental validation. While I go well along with the offered explanations, the authors could add at least two more limitations to the analysis. This is the PRS is more like the domain-domain AF data,

or at least theoretically the new yeast data are not as reliable. What is the average surface area of RPS pairs etc ...

AUTHOR RESPONSE

We thank the reviewer for making this important point. The average interface area of the subset of scPRS-v2 pairs for which there are experimental structures available is indeed higher than the full set of experimentally resolved PPIs (see plot below). There are 31 pairs of the scPRS-v2 in the 3DID domain-domain dataset (1,250 total DDIs) and only 1 scPRS-v2 pair in the ELM domain-motif dataset (95 total DMIs).

Interface area of PPIs in the yeast positive reference set, relative to other structurally resolved PPIs.

We do not believe the available evidence supports the hypothesis that the difference in confident structure prediction rates between YeRI and scPRS-v2 is primarily due to the reliability of the YeRI data. Explaining the observed discrepancy would require a false positive rate of ~60%, which is inconsistent with the validation results from the MAPPIT and GPCA assays. We have added the following to the main text:

EXCERPT FROM REVISED MANUSCRIPT

“The lower proportion of successful AlphaFold2 predictions in the novel YeRI pairs relative to scPRS-v2 could be due to a greater proportion of transient interactions, interactions involving disordered regions, and interactions between less evolutionarily conserved proteins detected in YeRI compared to the reference set⁴², in addition we noticed a bias in the scPRS-v2, which contains PPIs with significantly larger interface sizes on average ($P = 1.7 \times 10^{-6}$, two-sided Mann-Whitney U test).”

c) After detailed analysis of yeast data (figs 1-3) the authors perform the analyses of human data, with analogous outcome. While these results are very well supported and thus convincing, the human time line is not. In analogy, the human time line (Figure 4d) should include (similar to yeast Figure 3i), also the non-Vidal-Lab large scale data sets that contributed to the binary interactome mapping. The Vidal lab contributed the most significant data sets, but not exclusively. Including the most significant data sets also from other labs will strengthen the point of the Figure.

AUTHOR RESPONSE

We thank the reviewer for pointing this out and offer our apologies for this oversight. We have edited the timeline of reported human PPIs to include two other large-scale systematic binary datasets that we had previously missed (as shown below) and added the corresponding references to the text. Those datasets, MDC-HI1 (Stelzl *et al. Cell* 2005, with the full-length subset validated in Venkatesan *et al. Nature Methods* 2009) and NN1.0 (Haenig *et al. Cell Reports* 2020), were the only two we found, after searching the literature, that met our criteria for being validated using random samples in orthogonal assays and benchmarked using positive and random reference sets. If the reviewer is aware of any other datasets that meet these criteria, we will gladly add them to the timeline.

“36. Stelzl, U. *et al.* A Human Protein-Protein Interaction Network: A Resource for Annotating the Proteome. *Cell* 122, 957–968 (2005).

38. Haenig, C. *et al.* Interactome mapping provides a network of neurodegenerative disease proteins and uncovers widespread protein aggregation in affected brains. *Cell Rep.* 32, 108050 (2020).”

d) It should be noted that while I strongly agree that the novel PPIs in the human AF paper (which I have read carefully) are very few in number, they mostly relate to membrane-membrane PPIs, where Y2H is negatively biased. Therefore, the novel AF PPIs could be largely complementary to the experimental binary PPIs. This needs to be checked e.g. in terms of which proteins are involved etc. How many of the strictly novel PPIs of the human AF/RF core have been retested with Y2H-v4 in the presented experiments? I suggest a similar panel like the ones in Figure 4a that show the success of *de novo* screens would be revealing if shown for strictly novel PPIs (provided that they were tested).

AUTHOR RESPONSE

This is an interesting point. According to UniProt annotations, 14% (2,575/17,849) of the human AF/RF dataset is between two transmembrane proteins, and 12% (2,119/17,849) is between a transmembrane protein and a non-transmembrane protein. The corresponding fractions for the strictly novel pairs increase to 21% (1,028/4,789) and 18% (847/4,789), respectively. To test the impact of this on our results, we therefore generated Fig. 3a after removing PPIs involving transmembrane proteins. As can be seen, the trend, and hence our conclusions, remain the same:

Thus, the increased number of interactions involving membrane proteins is not responsible for the observed drop in Y2H recovery rate at lower contact probability scores.

Regarding the reviewer's questions about the number of tested PPIs: the number of strictly novel human predicted PPIs successfully tested in Y2H was 604, of which 102 were between two transmembrane proteins, 139 were interactions between transmembrane and non-transmembrane proteins, and 363 didn't involve a transmembrane protein. A version of the *de novo* screen results of Fig. 4a, restricted to

strictly novel predictions, with and without excluding transmembrane proteins is shown below:

It is apparent that the novel pairs test positive at very low rates and that this is not explained by the enrichment in PPIs involving transmembrane proteins. One possible explanation would be that the novel pairs are indeed of low quality. However, it also cannot be excluded that these pairs are enriched in other properties that are indeed complementary to the detection profile of Y2H, akin to interactions involving transmembrane domains. This question needs to be addressed in subsequent experimental and computational analyses that are beyond the scope of this manuscript.

e) *The predictions of novel Y2H were done with success even though the pairs were not in the AF predicted data set (maybe different scoring, ccc). However, where these novel pairs considered in the AF predictions in the first place? Did they end up below cut off (cf . Figure 4c). The human predictions were guided by known pairs though ...*

AUTHOR RESPONSE

Of the yeast AF/RF pairs below the '-core' cutoff, there are 17 in YeRI, of which 12 pass the confident contacts count (CCC) threshold, of which 5 already have an existing experimental structure and 1 has a homologous structure, leaving 6 pairs that were excluded from the 237 pairs based on their appearance in the AF/RF dataset below the 0.95 contact probability threshold. So, only a small contribution.

Under our definition of the 237 confidently modeled YeRI pairs that did not have previous structural models, we excluded all pairs in the full AF/RF dataset before the cutoff. We have clarified this by adding a section to the methods:

EXCERPT FROM REVISED MANUSCRIPT

“PPIs with no structural model

Yeast PPIs with no structural model were defined as not being present in I3D-exp-24, not having an interolog structure (defined above), and not being in the full AF/RF dataset (i.e., the dataset before applying the higher, experimentally derived contact probability threshold).”

To answer the question of why the novel confidently modeled novel YeRI pairs were not in the AF/RF dataset in more detail, we also looked at the search space that was screened in the AF/RF dataset and added our findings to the discussion:

EXCERPT FROM REVISED MANUSCRIPT

“For YeRI, AlphaFold could infer new structural models for 237 PPIs, of which 72% (171) were not in the AF/RF search space because the pair had few homologues (shallow MSA depth), whereas 28% (66) were tested and were below the thresholds. These could either have been missed by the screening algorithm whilst being detected by the experimental screening or could have been recovered due to the improved CCC interaction score for AlphaFold structures, which we developed based on the scPRS-v2/scRRS-v2 benchmarking data, and which outperformed the previously described metrics.”

Reviewer #2 (Remarks to the Author)

The manuscript by Lambourne et al consists of three parts. In the first, the authors assess the current state of interactome mapping for budding yeast. Using three binary interaction assays, they test subsets of pairs from various interaction datasets to evaluate their quality. Most of the interactions are retested using an updated version of the Vidal lab yeast two-hybrid (Y2H) pipeline that is able to positively identify more interactions (higher sensitivity) while maintaining a low false-positive rate. The main conclusion is that most datasets are of low quality in terms of accurately reporting direct protein-protein interactions, possibly in part because many datasets report indirect associations (i.e. proteins in a larger complex) in the form of interacting pairs. The authors combine the high-quality datasets into a single dataset they term ValBin-24 for validated binary interactions in 2024. This section provides a thorough and thoughtful overview of the current state of interactome mapping.

In the second part of the manuscript, the authors experimentally expand the yeast binary interaction map using i) their updated Y2H pipeline, and ii) a near complete yeast

ORFeome collection generated here by the authors (adding 921 ORFs to an existing collection of 4933 ORFs). This adds 1723 interactions to previous Y2H high confidence interactions, for a total of 4556 interactions. Both the expanded ORFeome collection and new interactions are of high value to the scientific community.

In the third part, the authors assess the accuracy of yeast and human protein interactions predicted by new structure prediction algorithms. They do this by testing all predicted yeast interactions and a random sampling of human protein interactions in their Y2H pipeline. I found this an interesting analysis to read. The main conclusion is that while predictions are of high quality, very few new interactions are predicted. The likely reasons for this are that structural prediction algorithms are better at prediction structures for proteins similar to proteins with known structures, and are not good at identifying interactions with disordered regions, which mediate many protein interactions. While these observations are not necessarily unexpected, it is very useful to have a thorough examination of the predictive power of new AI methods.

Overall, the work provides an advancement in our knowledge of protein-interactions and in tools (Y2H system and new ORFeome dataset). The evaluation of AI methods should be of broad interest and is an important step in assessing the use cases for these new tools.

I have little if any qualms with the experimental approaches used. The Vidal lab are absolute experts in the field of binary protein interaction mapping, having honed their Y2H pipeline over 2 decades of work and having increasingly emphasized the value of rigorous benchmarking of interaction assays with positive and negative reference sets. The comparisons with AlphaFold predictions also seem to have been thoroughly performed, with attention to the computational methods and thresholds used. My comments for improvement are below, most address readability.

AUTHOR RESPONSE

We thank the reviewer for this positive summary of our work.

- A table with all the various datasets mentioned would be very helpful, even being familiar with past work from the Vidal lab the many dataset abbreviations confused me. Most are explained in the methods, but an overview table to refer to while reading would be a welcome addition.

AUTHOR RESPONSE

We thank the reviewer for this useful suggestion. We have added both a new Extended Data Figure 3 and Tables 1 and 2, copied below:

Extended Data Figure 3: Schematic of the relationships between yeast datasets used in this study.

Name	Description	# heterodimer edges	# proteins
I3D-exp-17	Experimental PPI Structures	985	730
I3D-exp-24	Experimental PPI Structures	2,576	1,408
Lit-CC-17	Literature-curated co-complex	70,492	5,600
Lit-CC-23	Literature-curated co-complex	125,926	5,823
Lit-BS-17	Literature-curated binary single evidence	13,736	4,712
Lit-BS-23	Literature-curated binary single evidence	14,846	4,607
Lit-BM-13	Literature-curated binary multiple evidence	3,778	2,259
Lit-BM-17	Literature-curated binary multiple evidence	4,260	2,519
Lit-BM-23	Literature-curated binary multiple evidence	5,522	2,799
Uetz-screen	Systematic binary	607	747
Ito-core	Systematic binary	738	766
DHFR PCA	Systematic binary	2,525	1,076
CCSB-Y11	Systematic binary	1,605	1,206
Gavin (a)	Systematic AP/MS	3,210	1,352
Ho	Systematic AP/MS	3,583	1,555
Krogan	Systematic AP/MS	7,055	2,670
Gavin (b)	Systematic AP/MS	6,531	1,430
YeRI	Systematic binary	1,880	1,346
AF/RF	AI predictions	1,499	1,566
AF/RF-core	AI predictions	969	1,177

Table 1: Descriptions of the biophysical datasets used in this study. Dataset names ending in numbers refer to the year of generation.

Name	Constituent datasets	# heterodimer edges	# proteins
Y2H-union-08	Uetz-screen + Ito-core + CCSB-Y11	2,613	1,884
Y2H-union-25	Y2H-union-08 + YeRI	4,307	2,505
ValBin-24	I3D-exp-24 + Lit-BM-23 + Y2H-union-08	9,215	3,594
ValBin-25	ValBin-24 + YeRI + AF/RF-core	11,031	3,976

Table 2: Composition of composite PPI datasets.

- *The multiple Lit-BM sets are not well explained. On page 4 line 28, the authors state that interactions from Lit-BM-17 were tested, but on line 32 it seems a Lit-BM-13 was also tested. What is the difference?*

AUTHOR RESPONSE

The Lit-BM dataset is continuously updated every few years, and we performed each experiment with the most recent version at that time. The MAPPIT and GPCA experiments were performed with Lit-BM-13, whereas the Y2H was performed with Lit-BM-17. The datasets are largely overlapping, as you would expect. We have clarified this in the text as follows:

EXCERPT FROM REVISED MANUSCRIPT

“Specifically, we tested 8,999 pairs in Y2H-v4, which included all protein pairs with experimental 3D structures curated in 2017 (I3D-exp-17), literature pairs with multiple evidence of which at least one can detect binary PPIs (Lit-BM-17), and three systematically generated binary maps (Y2H-union-08) (Fig. 1f and Extended Data Table 10). Since most of the systematic binary approaches and many of the studies that comprise Lit-BM used the Y2H assay, we also tested Lit-BM-13, Y2H-union-08, and DHFR-PCA in GPCA and MAPPIT (Fig. 1g, h, and Extended Data Table 11). Of all datasets tested, we found that I3D-exp-17, Lit-BM-13/-17, and Y2H-union-08 tested comparably to the positive control set, whereas the other datasets, which make up the majority of the published pairs, tested at rates closer to the random pairs.”

- *On page 5 line 26, why are the new interactions compared to Lit-BM-13? In figure 1, Lit-BM-17 was used.*

AUTHOR RESPONSE

As stated for the previous point, the MAPPIT and GPCA experiments were performed using pairs originally identified in Lit-BM-13.

- Page 5 line 36: the sentence ‘thus we generated a new high quality map’ would be better placed on line 23, before going on to evaluating the new map. The way it reads now I was wondering if the new map was generated using criteria based on the comparison with the PSNs, e.g. only incorporating those that were enriched in one or more PSNs. (or if that is actually the case, clarify the criteria).

AUTHOR RESPONSE

We have moved the sentence.

- The naming of the Yeast Reference Interactome seems an odd choice. What makes this a reference interactome, why would CCSB-YI1 not be a reference interactome, or the next interactome the Vidal lab generates. Wouldn't a year name be more consistent with their nomenclature, e.g. CCSB-YI25? Or YI2? The analyses below are also not done on YeRI but on union-25 and on ValBin-25, so it isn't used as a reference set here either.

AUTHOR RESPONSE

We use the term reference interactome because of its all-by-all testing of the search space, i.e., close to a complete ORFeome collection. This is consistent with the naming of our human reference interactome, HuRI (Luck *et al.* *Nature* 2020). Previous systematic maps in yeast left substantial portions of the search space unscreened (see Fig. 2a below). By increasing the search space coverage to the point where it is effectively complete, YeRI takes this important variable out of consideration. The resulting gateway-compatible clone collection can then be employed by others, using different PPI assays, to build towards a complete map of the interactome (see Choi *et al.* *Nature Communications* 2019), with YeRI acting as a cornerstone of those future efforts. Importantly, the use of the term reference is analogous to that of the human reference genome, which, in its original form, was substantially incomplete but nevertheless did serve its valuable purpose as a reference for the community.

- The combining of YeRI with ‘the datasets previously identified as high-quality binary interactions’ wasn't clear to me. Do the authors mean ValBin-24? I also don't think that a table of ValBin-25 with the source(s) of each interaction is included.

AUTHOR RESPONSE

Yes, we were referring to ValBin-24. We thank the reviewer for highlighting that this was unclear. We have also added AF/RF-core to ValBin-25, since it is also a validated dataset of binary PPIs. We had originally included a table of ValBin-25, but by mistake, omitted the name in the description. We have added that in.

EXCERPTS FROM REVISED MANUSCRIPT

“Adding YeRI to the datasets we previously identified as high-quality binary interactions (ValBin-24) results in a total of 10,759 PPIs.”

“**Extended Data Table 22.** ValBin-25: the union of high-quality binary experimental and computational interactome maps. The union of I3D-exp-24, Lit-BM-23, AF/RF-core, and Y2H-union-25. Note: homodimers are included for completeness but were generally not used in the analysis in the manuscript.”

- On page 5, the authors note that the Y2H systems stands out for having contributed large numbers of reliable and primarily direct PPIs. If the authors want to make a point that Y2H is a better system than others, this statement needs more substantiation. Specifically, I think there may be a circular argument here for 2 reasons: 1) most available binary interactions were identified by Y2H, and 2) in the retesting, most interactions were retested by Y2H thus having a higher chance to retest positively than interactions identified by other methods. While most high quality direct interactions may stem from the Y2H system, this may therefore not be of particular note and not reflect an intrinsic higher quality of the Y2H system.

AUTHOR RESPONSE

This is a fair comment. We note that we also performed MAPPIT and GPCA assays to evaluate the quality of existing datasets. However, that statement, as it was written, did not accurately express what we wanted to say. We were not trying to focus so much on relative quality compared to other methods, but rather on its relative practicality, having been used to generate large high-quality datasets. So we have rephrased as follows:

EXCERPT FROM REVISED MANUSCRIPT

“Notably, of all the experimental methods, systematic high-throughput Y2H-based mapping has proven to be a particularly practical and scalable approach, having produced multiple large interactome maps of reliable and primarily direct PPIs.”

- The guilt-by-association experiments are difficult to interpret in their current form. The example in note 2 is clear, but the experiment is not. I understand that the authors first assign a potential function based on potential functions of its neighbour (GO term based).

Then, during the time between the analysis and paper writing, some proteins went from unknown to studied, allowing a determination of whether that assessment was correct. So I was expecting table 18 to have a list of proteins, a predicted function, and an assessment whether new information shows if that prediction is right. But that is now what table 18 shows. How should the reader interpret that table? Their example YGR168C is listed 86 times, so apparently 86 possible functions were assigned? And how do I see from the table that that assignment was correct?

AUTHOR RESPONSE

Thank you for pointing out this lack of clarity. We have updated the column headings and the table description.

In the table, the 86 entries for YGR168C are each different GO terms that exceeded the Z-score > 2 threshold, for both networks the predictions were performed on (YeRI and Y2H-union-25). The main reason there are so many is because of the large redundancy in GO terms. For example, the top three terms (ranked by p-value) for that gene are: GO:0005778 peroxisomal membrane, GO:0007031 peroxisome organization, and GO:0016559 peroxisome fission. Another reason for the large number of entries is because we used a very relaxed threshold of Z-score > 2 when generating the table, with the idea that the reader would rank and filter themselves. On reflection, this was a poor choice, and so we have increased the stringency of the predictions in the table – to Z-score > 5 and at least 2 partners with the predicted GO term – which has reduced the number of rows in the updated table.

Regarding the genes whose names had changed, previously we had simply inspected the top handful of predictions and found that the top prediction, YGR168C, had been renamed PEX35. We have now added comments on the other 3 predictions which have been renamed to the text.

EXCERPT FROM REVISED MANUSCRIPT

“There are four such examples, which can be used to test the accuracy of our predictions. Firstly, owing to its recently demonstrated role as a regulator of peroxisomal abundance, YGR168C is now known as PEX35⁶⁸. Ygr168c/Pex35 has 23 PPIs in Y2H-union-25, all from YeRI, out of which eight are proteins involved in peroxisomal biology, and so was predicted to function in peroxisomal protein import, showcasing the ability of Y2H-union-25 to predict gene function accurately (Extended Data Fig. 3i). Moreover, Pex35 interacts with another protein of unknown function, Ykl018c-a/Mco12, which can be now hypothesized to be involved in peroxisomal abundance as well. Secondly, the highest-ranked prediction for Yjr011c was to be involved in the CCR4-NOT complex, and it is now known as Cal4, an accessory component of the CCR4-NOT complex⁶⁹. Thirdly, the

highest-ranked prediction for Ykl075c is for mitochondrial respiratory chain complex II assembly. This gene is now named AAN1, based on “impacts on actin cable stability, mitochondrial function, BCAA metabolism, and cellular lifespan”, with its deletion being found to significantly reduce mitochondrial oxidation levels⁷⁰. Ykl075c also localizes to the mitochondria upon rapamycin treatment⁷¹. Finally, Ypr174c is predicted to be localized to a membrane. It is now named Csa1 and found to anchor Cdc5 at spindle pole bodies⁷². It binds phosphatidylinositols and phosphatidylethanolamines⁷³ and is localized to the nuclear periphery⁴⁵. In summary, where there is evidence to evaluate our predictions, it broadly supports their accuracy, especially considering the heuristic nature of their generation, and thus supports the useful information contained within our systematically generated PPI network.”

69. Pillet, B. *et al.* Dedicated chaperones coordinate co-translational regulation of ribosomal protein production with ribosome assembly to preserve proteostasis. *Elife* 11, (2022).

70. Sing, C. N. *et al.* Identification of a modulator of the actin cytoskeleton, mitochondria, nutrient metabolism and lifespan in yeast. *Nat. Commun.* 13, 2706 (2022).

71. Koh, J. L. Y. *et al.* CYCLOPs: a comprehensive database constructed from automated analysis of protein abundance and subcellular localization patterns in *Saccharomyces cerevisiae*. *G3: Genes|Genomes|Genetics* 5, 1223–1232 (2015).

72. Örd, M. *et al.* Proline-rich motifs control G2-CDK target phosphorylation and priming an anchoring protein for Polo kinase localization. *Cell Rep.* 31, 107757 (2020).

73. Gallego, O. *et al.* A systematic screen for protein-lipid interactions in *Saccharomyces cerevisiae*. *Mol. Syst. Biol.* 6, 430 (2010).”

- Page 6 line 19 ‘testing all 1505 pairs’. This sounds like a new experiment, but this is just data taken from YeRI correct? (Since that experiment tested all pairwise combinations).

AUTHOR RESPONSE

No, this is a new experiment. YeRI was not generated by testing each pair individually, since that number is not experimentally feasible. Rather, we first tested individual DB-ORF fusions with pools of hundreds of AD-ORF fusions, referred to as the screens, and then subsequently pairwise tested the hits from the screens. The experiments with all 1,505 AF/RF yeast pairs were pairwise tests, which we performed since they have significantly higher sensitivity than the pooled screens. We have clarified this in the text as follows:

EXCERPT FROM REVISED MANUSCRIPT

“To assess the accuracy of these predictions, we experimentally evaluated the yeast AF/RF dataset, performing Y2H-v4 pairwise tests of all 1,505 pairs.”

- Page 7 line 11: how do the strictly novel predicted PPIs fare in the Y2H retest?

AUTHOR RESPONSE

There are 42 strictly novel predicted pairs, of which 30 were successfully tested (after excluding autoactivators and experimental failures). None tested positive when benchmarked against a PRS/RRS set. As in response to a similar question by Reviewer 1, it is possible that the newly predicted pairs represent high-scoring false positives of the AF/RF prediction. At the same time, it must be kept in mind that these are a relatively small number of pairs, thus limiting the informativeness of the data. It is possible that the predicted pairs have properties that bias them against detection by our validation assay, such as the increased proportion of pairs involving one or more transmembrane proteins, which are detected at lower rates in Y2H. This question must be resolved in future studies.

- Page 7 line 39: why was Y2H-v1 used to retest the predicted human interactions?

AUTHOR RESPONSE

For practical reasons and to perform the experiment as quickly as possible. We already had the entire human ORFeome in the v1 vectors, but not in v4.

- Page 9 line 38: shouldn't ValBin-25 be the new gold-standard?

AUTHOR RESPONSE

Yes. We thank the reviewer for spotting this mistake and have corrected it in the text.

- Page 10 line 14/15. If AlphaFold can predict models for interactions not detected in the interaction screen, does that mean that the computational prediction pipeline can be improved and identify more of the missing interactions?

AUTHOR RESPONSE

We believe this is the case, yes. In our view, key points for improvement would be to (a) increase the sensitivity of the fast screening algorithm to approach that of AlphaFold; (b) increase the search space coverage by removing the filters; (c) use an improved scoring

metric, i.e., the confident contacts count (CCC). Importantly, however, the potential gains of these improvements to the fast screening algorithm would still be constrained to an upper limit of the sensitivity of the slower AlphaFold, which we estimate to be 14%, based on running AlphaFold on the YeRI PPI dataset.

- Page 10 line 20: typo in 'could be may have been'

AUTHOR RESPONSE

We have corrected it in the text.

- *Supplementary info: in generating the new negative reference set the authors filter out any pair that appeared in any experimental dataset. While a logical approach, I wonder if at some point you start to bias the negative reference set for protein pairs that are particularly unlikely to be detected as false positives, for example because they have overall opposing charges or structures that are not likely to be 'sticky'. In theory you want to test true random protein pairs but this approach is no longer truly random. Perhaps only validated direct protein interactions should be subtracted.*

AUTHOR RESPONSE

This is a good point, which we will consider in future work. Whilst we checked for experimentally determined interactions when generating scRRS-v2 (random reference set), we did not find any that were picked as part of the random selection, and hence, no randomly picked pair was actually removed. In principle, though, the reviewer raises a valid point, and it's something we will consider going forward. In general, the generation of negative interaction datasets that are large, clean, and unbiased is fraught with challenges.

Reviewer #3 (Remarks to the Author):

On the manuscript entitled "Experimental assessment of AI-based interactome mapping", Vidal and co-authors present a comprehensive effort to evaluate the performance of AlphaFold in global deciphering of quaternary structures (protein-protein interactions, PPIs) in yeast and human, and at the same time provide the community with a new experimental method for detection of such interactions that provide a large number of novel high-quality PPIs. The work is thorough and sound. The presented study is endowed with multiple virtues and contributions, among which the critical assessment of the usability of PPI data is not the least. Overall, the raised claims, the presented results and the contextual discussion correspond well with each other. The manuscript is well-written (although sometimes a bit difficult to read, due to the lack of space and the need

to consult with the profuse additional materials). In my opinion, the manuscript would be of the highest interest in PPI research and may also appeal to the wider community, and has my enthusiastic support for its publication in Nature Communications.

AUTHOR RESPONSE

We thank the reviewer for this positive summary.

There are a few aspects that, in my opinion, could make the manuscript stronger, however. I will list them below for the Author's consideration:

1. One of the main contributions of the manuscript (highlighted at the very end of the discussion) is the new YeRI dataset, providing over 1400 novel protein interactions in yeast. This is achieved by implementing a variation of the Yeast-two-Hybrid (Y2H) complementation assay, aptly named Y2H-v4 in the manuscript. For the benefit of the wider audience, I think the manuscript would benefit from a more thorough explanation of the technique and an explanation of its similarities and differences with the previous versions of Y2H. Ideally, Fig.1 (and failing to that, the corresponding supplementary information) could allocate these explanations.

AUTHOR RESPONSE

We thank the reviewer for highlighting this contribution, and we've made changes to make the differences between Y2H-v4 and Y2H-v1 clearer in the paper.

We've added a new Extended Data Figure 1e as a schematic overview of the Y2H v1 vs v4 differences:

e

We've added a new Methods Table 3 to make the assay differences clearer:

Methods Table 3. Y2H assay version comparison

Assay version	DB vector	AD vector	DB yeast strain	AD yeast strain
1	pDEST-DB	pDEST-AD-CYH 2	Y8930	Y8800
4	pDEST-DB-QZ212	pDEST-AD-QZ213	Y8930	Y8800

Methods Table 4. Yeast destination vectors

Name	pDEST-DB	pDEST-DB-QZ212	pDEST-AD-CYH2	pDEST-AD-QZ213
Fusion partner (aa)	Gal4-DB (1-147)	Gal4-DB (1-147)	Gal4-AD (768-881)	Gal4-AD (768-881)
Fusion location	N-terminus	N-terminus	N-terminus	N-terminus
Yeast Promoter (nt)	Truncated ADHI (-701 to +1)	Truncated ADHI (-410 to +1)	Truncated ADHI (-701 to +1)	Truncated ADHI (-410 to +1)
Yeast replication of origin	CEN	2 μ	CEN	2 μ
Linker sequence between C-Term of Gal4 element and Gateway cloning site (aa)	SRSNQ	PEFPS	GGSNQ	ICMAYPYDVPDYASLGGHM AMEAPS
Yeast terminator	ADHI Term	ADHI Term	ADHI Term	ADHI Term
E. coli selection marker	Ampicillin	Ampicillin	Ampicillin	Ampicillin
Yeast auxotrophic selection marker	LEU2	LEU2	TRP1	TRP1

2. The Authors mention a large number of existing datasets, especially in the first part of the manuscript (yeast). A graphic navigating the reader through those datasets would be extremely useful, either in the main manuscript or in its accompanying supplementary information. Such a guide could include the nature of each of the sets, their size, and the reason for being used in the current study.

AUTHOR RESPONSE

We thank the reviewer for this suggestion and have added new Extended Data Figure 3 and Tables 1 and 2.

Extended Data Figure 3: Schematic of the relationships between yeast datasets used in this study.

Name	Description	# heterodimer edges	# proteins
I3D-exp-17	Experimental PPI Structures	985	730
I3D-exp-24	Experimental PPI Structures	2,576	1,408
Lit-CC-17	Literature-curated co-complex	70,492	5,600
Lit-CC-23	Literature-curated co-complex	125,926	5,823
Lit-BS-17	Literature-curated binary single evidence	13,736	4,712
Lit-BS-23	Literature-curated binary single evidence	14,846	4,607
Lit-BM-13	Literature-curated binary multiple evidence	3,778	2,259
Lit-BM-17	Literature-curated binary multiple evidence	4,260	2,519
Lit-BM-23	Literature-curated binary multiple evidence	5,522	2,799
Uetz-screen	Systematic binary	607	747
Ito-core	Systematic binary	738	766
DHFR PCA	Systematic binary	2,525	1,076
CCSB-Y11	Systematic binary	1,605	1,206
Gavin (a)	Systematic AP/MS	3,210	1,352
Ho	Systematic AP/MS	3,583	1,555
Krogan	Systematic AP/MS	7,055	2,670
Gavin (b)	Systematic AP/MS	6,531	1,430
YeRI	Systematic binary	1,880	1,346
AF/RF	AI predictions	1,499	1,566
AF/RF-core	AI predictions	969	1,177

Table 1: Descriptions of the biophysical datasets used in this study. Dataset names ending in numbers refer to the year of generation.

Name	Constituent datasets	# heterodimer edges	# proteins
Y2H-union-08	Uetz-screen + Ito-core + CCSB-Y11	2,613	1,884
Y2H-union-25	Y2H-union-08 + YeRI	4,307	2,505
ValBin-24	I3D-exp-24 + Lit-BM-23 + Y2H-union-08	9,215	3,594
ValBin-25	ValBin-24 + YeRI + AF/RF-core	11,031	3,976

Table 2: Composition of composite PPI datasets.

3. Following the previous point, the Authors define Y2H-union-25 as the merging of their YeRI with three previously existing datasets, comprising a total of 4,307 PPIs. I had issues identifying which previously existing datasets the Authors refer to. I guess these are refs. 11, 18, and 19, but clarification on the manuscript would be nice. On the same tone, on pg. 6, the Authors state that adding YeRI information to “previously identified as high-quality binary interactions results in a ValBin-25 dataset of 10,759 PPIs.” It would be valuable if the Authors could identify (refer to) those previously described high-quality binary interactions.

AUTHOR RESPONSE

We have edited the sentences to now read as follows:

EXCERPTS FROM REVISED MANUSCRIPT

“Therefore, we combined YeRI with the three previous systematic Y2H maps, Uetz-screen¹⁸, Ito-core¹⁹, and CCSB-Y11¹¹, resulting in Y2H-union-25 comprising 4,307 PPIs (Extended Data Table 17).”

“Adding YeRI to the datasets we previously identified as high-quality binary interactions (ValBin-24) results in a total of 10,759 PPIs.”

4. A key question that is only partially addressed during the text is the true complementarity or orthogonality nature of different methods. I will illustrate this with the following case. YeRI detects around 1900 PPIs (of which more than 1400 novel) while the estimates for the total yeast interactome range between 20 and 40 thousand (pg. 3). The Authors, later by integrating YeRI to Y2H-union24 and further to ValBin25, increase the coverage to around 10 thousand. The open questions here are twofold: what are the remaining (10-30K) yeast PPIs? How can a naive user remove older datasets containing incorrectly recorded PPIs? The Authors address the latter question in the second part of the manuscript (human), stating that various thresholds on different estimation techniques

can be applied (Fig. 4a), but a clearer guideline on what can be considered a wrong PPI would be useful to the readers.

AUTHOR RESPONSE

It is not clear what exactly is specific about the missing PPIs, but we can speculate on some possible characteristics of the large number of PPIs that remain to be mapped. It's likely that undetected PPIs are, on average, weaker than detected PPIs. They may also be more likely to require post-translational modifications to one or other of the two proteins or require a specific context in other ways. They may involve proteins that are more challenging experimentally, such as membrane proteins or very large proteins.

In general, the question of what the remaining PPIs to be mapped is very interesting but too broad to be included in the current manuscript. But we're currently preparing a separate manuscript, looking at the difference between stable co-complex and transient outside complex interactions, that at least partially addresses this question.

For the second question, the naive reader can use the ValBin-25 dataset as a high-confidence source of interactions in yeast. In generating that dataset, we have removed datasets that didn't meet our quality cutoff for direct interactions.

5. *On page number 8, the Authors state that the human AF/RF-core dataset consists of 87% of the predicted PPIs. This number is (strikingly?) different to that obtained for yeast (64%, pg. 6). Could the Authors comment if these differences are significant and, if so, what are their plausible origins?*

AUTHOR RESPONSE

This difference is significant, and we assume that this reflects improvements from the authors in their methodology for the generation of the human dataset from the yeast one. The key difference is splitting the single contact probability threshold into multiple different thresholds and setting lower thresholds for pairs with previous evidence (in PPI databases or genetic interactions) and using more stringent thresholds for hits from their *de novo* computational screen. This seems to have increased the dataset's precision, but without increasing the novelty. They have also changed their screening algorithm, which seems to have been an improvement, but it is impossible to say with certainty, as the differences could also be largely as a result of the switch from yeast to human.

6. *On the side of minor notes:*

6.1. *At the beginning of the manuscript, the Authors refer systematically to the positive reference set as scPRS-v2. As the manuscript develops, this is sometimes substituted by*

PRS. Unity on naming would help the reader to understand that the Authors are referring to the same dataset.

AUTHOR RESPONSE

We have fixed this inconsistency in the revised version of the text.

6.2. Despite the amazing effort in keeping the text short and clear, I think the Authors have some editing errors on page 10, line 20. The text reads “[...] some of the interactions for which structures could be may have been missed [...]”. I think the current grammatical formulation is incorrect. Instead, “could have been missed” or “may have been missed” sounds better.

AUTHOR RESPONSE

Thank you for the kind words and for pointing out this mistake. We have fixed it in the revised version.